



# Using variable-resolution grids to model precipitation from atmospheric rivers around the Greenland ice sheet

Annelise Waling[1], Adam Herrington[2], Katharine Duderstadt[1], Jack Dibb[1], and Elizabeth Burakowski[1]

[1]Institute for the Study of Earth, Oceans, and Space, University of New Hampshire, Durham, NH, USA
[2]National Center for Atmospheric Research, 1850 Table Mesa Drive, Boulder, Colorado, USA

**Correspondence:** Annelise Waling (annelisewaling@gmail.com)

**Abstract.** Atmospheric rivers (ARs) are synoptic-scale features that transport moisture poleward and have been shown to cause short duration, high-volume melt events on the Greenland ice sheet (GrIS). This project investigates the effectiveness of variable-resolution (VR) grids in modeling ARs and their subsequent precipitation around the GrIS using a study period of 1 January 1979 to 31 December 1998. VR simulations from the Community Earth System Model (CESM2.2) bridge the gap between limitations of global climate models and regional climate models while maximizing computational efficiency. VR grids improve the representation of ARs, in part by resolving small-scale processes. ARs from CESM2.2 simulations using three grid types (VR, latitude-longitude, and quasi-uniform) with varying resolutions are compared to output of ERA5 and MERRA2 observation-based reanalysis products.

The VR grids produce ARs with smaller areal extents and lower integrated precipitation over the GrIS compared to latitude-longitude and quasi-uniform grids. We hypothesize that the smaller areal extents in VR grids are produced by the refined topography resolved in these grids. In contrast, smoothing from coarser resolution latitude-longitude and quasi-uniform grids allow ARs to penetrate further inland on the GrIS. The reduced areal extent in VR grids also likely contributes to the lower area-integrated cumulative precipitation, whereas the area-average cumulative precipitation is similar for VR, latitude-longitude, and quasi-uniform grids. The VR grids most closely match the AR overlap extent and precipitation in ERA5 and MERRA2, suggesting the most realistic behavior among the three configurations.

## 1 Introduction

Atmospheric rivers (ARs) are large filamentary structures within the atmosphere that contain concentrated amounts of water vapor. ARs originate in the mid-latitudes and subsequently travel poleward. Nearly 90% of total annual polar moisture transport is attributed to ARs (Payne et al., 2020). In addition to bringing large amounts of water vapor to the poles, ARs often bring warm temperatures and contribute to snow and ice melt (Bonne et al., 2015; Mattingly et al., 2018, 2020; Box et al., 2022; Mattingly et al., 2023). Polar regions are sensitive to feedbacks and warming induced melting, and ARs can exacerbate extreme melting events (Payne et al., 2020). For example, in July 2012 the Greenland ice sheet (GrIS) experienced a short-duration, high-volume melt event in association with an AR, which caused substantial mass loss. Bonne et al. (2015) found that during this event, surface mass balance (SMB) fell three standard deviations below the average value during this time of year and



surface melt covered 97% of the GrIS. Before the 2012 event, the most recent instance of melt covering nearly the entire GrIS was 1889 (Neff et al., 2014).

Researchers have observed and predicted an increase in both frequency and intensity of ARs as climate change progresses (Lavers et al., 2015; Hagos et al., 2016; Espinoza et al., 2018; Curry et al., 2019; Gershunov et al., 2017; Huang et al., 2020; Zhang et al., 2021, 2023). This trend suggests that ARs impacting the GrIS SMB, such as the July 2012 event, will increase in frequency. The GrIS experienced another major melt event in mid-August 2021, which caused rainfall at Summit Station (Box et al., 2022).

Studies have been conducted regarding the effects of tracking algorithms on AR detection (Shields et al., 2018, 2023), but only a few studies have been published on the effect of grid configuration choice on AR modeling (Hagos et al., 2015). Within the modeling community there has been a shift in preferred grid configurations to rectify the "polar problem" where grid spacing in standard latitude-longitude grids decreases towards the poles. As grid spacing decreases, the numerics within the climate model become unstable and a dampening polar filter is necessary to prevent any numerical instability. Implementing such filters reduces some of the benefit of refined resolution towards the poles, though not entirely (Herrington et al., 2022). Some modelers have recently run simulations using quasi-uniform unstructured grids, e.g., the spectral-element (SE) grid dynamical core (Lauritzen et al., 2018). While these grids eliminate the need for a polar filter and allow for increased computing efficiency, they have coarser spatial resolution in polar regions. Variable-resolution grids may alleviate some of the negative effects of latitude-longitude schemes while enabling high spatial resolution in polar regions, though this comes at a higher computation cost.

We use variable-resolution grids (Zarzycki and Jablonowski, 2015; Zarzycki et al., 2015) with the Community Earth System Model version 2.2 (CESM2.2) (Danabasoglu et al., 2020; Herrington et al., 2022) to model ARs around the GrIS. VR grids employ static mesh refinement to yield enhanced resolution around our region of interest, Greenland. We compared VR-CESM2.2 model results to two quasi-uniform unstructured and two latitude-longitude grid simulations. We hypothesize that the VR grids will simulate ARs more accurately than the coarser resolution grids through better resolution of finer-scale physical processes and topography, as has been seen in other studies investigating moisture intrusions in the Arctic (Bresson et al., 2022). Accurately modeling precipitation from ARs is important because it has been suggested that during early summer nearly 40% of precipitation in Greenland is due to ARs (Lauer et al., 2023). The model output is compared to ARs detected by ERA5 and MERRA2, two observation-based meteorological reanalysis datasets, as in other studies involving simulated ARs (Bresson et al., 2022; Viceto et al., 2022; Zhou et al., 2022; Mattingly et al., 2023).

This study compares AR characteristics and precipitation from ARs produced by latitude-longitude, quasi-uniform, and variable-resolution grids in CESM2.2 Section 2 describes the model grids, remapping workflow, AR detection method, precipitation counting method, and the validation datasets used in this study. Section 3 contains the main results and analyses performed in this project. Section 4 discusses the implications of these results. Section 5 summarizes main conclusions from our work and provides direction for future research.



## 2 Methods

### 2.1 Model simulations

This study uses model output from the simulations described in Herrington et al. (2022) using CESM2.2 (Danabasoglu et al., 2020; Herrington et al., 2022), a CMIP6-class (Coupled Model Intercomparison Project Phase 6) (Eyring et al., 2016) Earth System Model. The simulations were configured with the Atmospheric Model Intercomparison Project (AMIP) protocols, which prescribe monthly sea-surface temperature and sea ice following Hurrell et al. (2008).

Herrington et al. (2022) ran CESM2.2 simulations at six different grid resolutions (Table 1, Figure 1) from 1 January 1979 to 31 December 1998. These include two latitude-longitude (LL) grids, two quasi-uniform unstructured grids (QU), and two variable-resolution (VR) grids. LL grid configurations use the finite-volume (FV) dynamical core (hereafter referred to as dycore), with a flux-form Lagrangian scheme (Lin and Rood, 1997) in the horizontal direction and semi-Lagrangian discretization in the vertical (Lin, 2014). The QU grids use spectral-element (SE) dycores and the Galerkin spectral finite element method (Fournier et al., 2004). These dynamics are solved with high-degree piecewise polynomials, and therefore have improved numerical accuracy in the horizontal compared with the FV dycore. SE uses the same semi-Langrangian vertical discretization as the FV dycore. SE dycores are ideal for high resolution modeling for improved computational efficiency on massively parallel systems, as well as including the effects of condensates that can greatly influence the dynamics of a system at high resolution (Bacmeister et al., 2012; Lauritzen et al., 2018). With their high computational efficiency, SE dycores also support VR grids, including the two presented in our study (Table 1).

**Table 1.** Description of grid configurations.

| grid name | grid type[a] | dynamical core | grid spacing[b] (°) | $\Delta x_{refine}$[c] (°) | ensemble members[d] |
|---|---|---|---|---|---|
| f19 | LL | FV | 2 | - | ESMF-pg2, TR-pg2, native |
| f09 | LL | FV | 1 | - | ESMF-f19, ESMF-pg2, TR-f19, TR-pg2 |
| ne30pg2 | QU | SE-CSLAM | 1[e] | - | ESMF-f19, TR-f19, native |
| ne30pg3 | QU | SE-CSLAM | 1 | - | ESMF-f19, ESMF-pg2, TR-f19, TR-pg2 |
| ARCTIC | VR | SE | 1 | 0.25 | ESMF-f19, ESMF-pg2, TR-f19, TR-pg2 |
| ARCTICGRIS | VR | SE | 1 | 0.125 | ESMF-f19, ESMF-pg2, TR-f19, TR-pg2 |
| ERA5 | - | - | 0.25 | - | ESMF-f19, ESMF-pg2, TR-f19, TR-pg2 |
| MERRA2 | - | - | 0.5x0.625 | - | ESMF-f19, ESMF-pg2, TR-f19, TR-pg2 |

**Table 1.** [a]LL = longitude-latitude, QU = quasi-uniform, VR = variable-resolution

[b]Average equatorial grid spacing.

[b]Grid refinement for variable resolution grids.

[d]Remappings performed which were included in final ensemble. ESMF-f19/TR-f19 and ESMF-pg2/TR-pg2 refer to ESMF and TempestRemap methods which transformed native grids to f19 and ne30pg2, respectively. Note that f19 and ne30pg2 grids were not remapped to themselves, their native grid configurations were used.

[e]While ne30pg2 has the same 1° spacing as ne30pg3, ne30pg2 has reduced physics resolution therefore degrading this 1° resolution



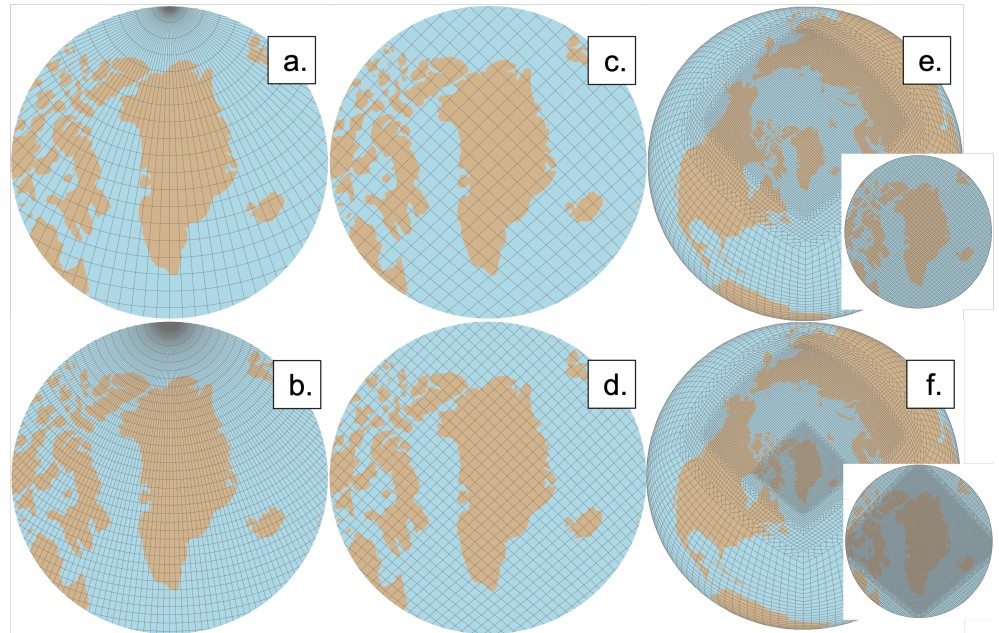

**Figure 1.** Grids used in this study. a-b) Latitude-longitude (LL) (a- f19, b- f09) grids with higher resolution in polar regions. c-d) Quasi-uniform (QU) (c- ne30pg2, d- ne30pg3) grids with more consistent resolution throughout the globe. e-f) Variable-resolution (VR) (e- ARCTIC, f- ARCTICGRIS) with insets emphasizing the higher resolution in the Arctic and Greenland. Lower resolution grids are shown on top row and high resolution on bottom row. Adapted from Herrington et al. (2022).

The atmosphere simulations used the Community Atmosphere Model 6.3 (CAM6) (Craig et al., 2021) with data recorded at six-hourly intervals. The variables used from CAM6 were convective precipitation rate (PRECC) and large-scale stable precipitation rate (PRECL), which were summed to reach the total atmospheric precipitation (PRECT). The ERA5 precipitation variable is also total precipitation (PRECT) and MERRA2 is the bias corrected total precipitation (PRECTOTCORR) The IVT fields from the CAM simulations were used in AR detection (uIVT, vIVT).

The Community Land Model 5.0 (CLM5; (Lawrence et al., 2019)) was coupled to CAM6 and provided daily averaged precipitation rate. Topography is a boundary condition for CAM (Danielson and Gesch, 2011). Software for processing this topography into CAM boundary conditions is attributed to (Lauritzen et al., 2015). In CLM5, grid cells are divided into multiple land unit types. For Greenland, this includes primarily the 'Glacier' and 'Vegetated/bare ground' land unit types. For our analysis, we are interested in AR interactions in the 'Glacier' land units on the GrIS, provided at 30-second resolution by Rastner et al. (2012), and exclude grid cells that are dominated by non-glacier land units. The high resolution GrIS mask was then aggregated to the model grid resolutions shown in 1.

Figure 2 shows the impact of grid configuration on the resolution of the topography in Greenland. In the coarser grid configurations (LL, QU), the elevation gradient from the coastal regions to the summit is not well represented. In addition to this, the high elevation in the middle of the GrIS is smoothed in the coarser grids, resulting in a flatter ice sheet.



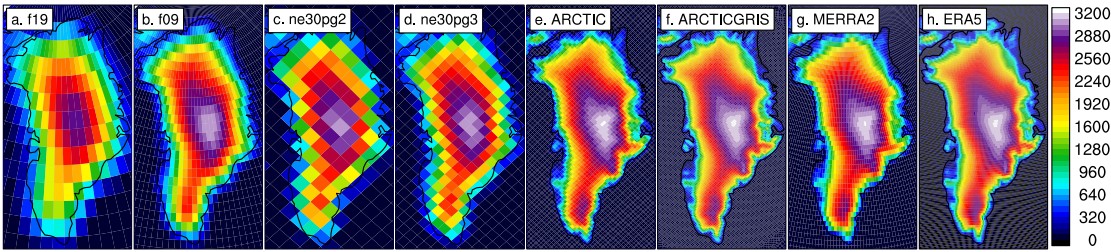

**Figure 2.** Topography of each grid configuration and reanalysis dataset used in this study, with higher resolution grids more accurately capturing the elevation gradients in Greenland. A-b show latitude-longitude (LL) (a- f19, b- f09) grids, c-d quasi-uniform (QU) (c- ne30pg2, d- ne30pg3), and e-f variable-resolution (VR) (e- ARCTIC, f- ARCTICGRIS), g shows MERRA2, and h shows ERA5.

### 2.1.1 Remapping

We remapped the simulations from each grid configuration to the coarsest LL grid (f19) and the coarsest QU grid (ne30pg2) using two remapping methods, thus resulting in four ensemble members. This was a conservative choice in order to favor up-scaling instead of down-scaling. The two remapping methods were ESMF (Team et al., 2021) and TempestRemap (Ullrich and Taylor, 2015). For each simulation, the algorithm to identify and track ARs described in section 2.3 was run four times, once for each ensemble member.

Remapping was conducted using the ncremap algorithm in the netCDF Operator open-source geospatial data analysis software (NCO) (Zender, 2008) and relevant weight files. These weight files describe the transformation from one grid configuration to either f19 or ne30pg2. In most cases, transforms are performed from fine resolution to coarser resolution, though when mapping from ne30pg2 to f19 down-scaling is taking place.

### 2.2 Detecting Atmospheric Rivers

Synoptic storms were tracked using TempestExtremes v2.1 atmospheric feature detection software (Ullrich et al., 2021). This algorithm was chosen to detect ARs due to its usage of the Laplacian of the integrated water vapor transport (IVT) rather than IVT alone. The gradients identified by the Laplacian method can detect ARs more accurately because there will still be a steep gradient between the AR itself and any surrounding moist area, thus better constraining the geometry of the AR (McClenny et al., 2020).

Though the threshold requiring a minimum value of the Laplacian of IVT in TempestExtremes allows for characteristic detection of the geometry of ARs, this method also allows for similar behavior at higher latitudes to be classified as ARs





(see Section 3.1). Previous studies have noted the challenges of detecting polar atmospheric rivers due to the east-westward
wind patterns that emerge (Rutz et al., 2019). There are many AR tracking algorithms that exhibit different behaviors and are
suited to tracking ARs in specific locations (Shields et al., 2018). For example, when detecting Antarctic ARs, trackers which
emphasize zonal IVT produce more accurate ARs than other algorithms (Shields et al., 2022). As our study focuses on the
impact of resolution on ARs, including a limited number of high latitude regions of moisture transport in the AR analysis does
not affect the results.

Two algorithms from the TempestExtremes v2.1 package were used to detect and track ARs: one for detecting ARs (DetectBlobs)
and one for stitching ARs together through multiple timesteps (StitchBlobs). The detection algorithm searches the global extent
for ARs meeting parameters: Laplacian of IVT $< -30,000 \, \mathrm{kg \, m^{-2} \, s^{-1} \, rad^{-2}}$, above 20° latitude, and areal extent 566,666 km$^2$.
The Laplacian IVT threshold was chosen based on Ullrich et al. (2021). They chose an IVT of -20,000 $\mathrm{kg \, m^{-2} \, s^{-1} \, rad^{-2}}$; our
threshold is stricter and requires a larger gradient. The areal extent was chosen conservatively as $\frac{2}{3}$ the area of an average AR,
which is 850,000 km$^2$ (A. Rhoades, 2022, personal communication). IVT is defined by,

$$IVT = \sqrt{uIVT^2 + vIVT^2} \qquad (1)$$

where uIVT and vIVT are pointwise vertically integrated zonal and meridional vapor transport, respectively.

The output of the detection algorithm is a binary mask outlining candidate ARs and the stitching algorithm is used to connect
the blobs in time, providing each AR its own unique identification number. The stitching algorithm links the ARs detected at
each timestep by the detection algorithm, rejecting candidate blobs that are not continuous in time. Using these two algorithms
together, we track a single AR across its entire lifespan, from its origin in the mid-latitude regions, poleward transport, and
eventual dissipation. We chose to run the stitching algorithm using standard default settings based on optimizations from A.
Rhoades (personal communication, 2022). The number of ARs varied based on whether the native grid was remapped to f19 or
ne30pg2 and on the remapping method (Table 2). The spread between ARs intersecting the GrIS remapped to f19 and ne30pg2
in TempestRemap allows for a larger amount of diversity in our ensemble.

## 2.3 Compositing variables

To analyze the effects of ARs on precipitation over the GrIS, we first found all ARs that intersect the GrIS at some point in
their lifetimes. We counted all ARs touching the 'Glacier' land units of Greenland in the CLM, determined the overlapping
area of these ARs at each timestep, and calculated precipitation within these areas.

For each ensemble member, the tracker produces a binary mask array $B_n^i(t)$, that contains 1's for times $t$ and grid columns $n$
where blob number $i$ is active, and 0's elsewhere. Note that there is only one horizontal dimension $n$, which is the convention
for unstructured grids; a second horizontal dimension needs to be added when applying these equations to LL grids, e.g.,
$B_{x,y}^i(t)$.



**Table 2.** Number of ARs intersecting the GrIS.

| grid name | ESMF | | | TempestRemap | | | |
|---|---|---|---|---|---|---|---|
| | f19 | ne30pg2 | $\Delta$[a] | f19 | ne30pg2 | $\Delta$[a] | average[b] |
| f19 | 381 | 339 | 42 | 381 | 281 | 100 | 346 |
| f09 | 431 | 420 | 11 | 510 | 356 | 154 | 429 |
| ne30pg2 | 474 | 485 | 11 | 632 | 485 | 227 | 499 |
| ne30pg3 | 483 | 447 | 36 | 596 | 458 | 138 | 496 |
| ARCTIC | 441 | 404 | 37 | 572 | 405 | 167 | 456 |
| ARCTICGRIS | 397 | 359 | 38 | 520 | 359 | 161 | 409 |
| | | | | | | | |
| ERA5 | 426 | 374 | 52 | 425 | 376 | 49 | 400 |
| MERRA2 | 517 | 467 | 50 | 519 | 472 | 47 | 494 |

**Table 2.** [a]Difference ($\Delta$) between f19 and ne30pg2 detected ARs intersecting GrIS for each remapping method.
[b]Average takes into account ESMF-f19, ESMF-ne30pg2, TempestRemap-f19, and TempestRemap-ne30pg2.

We seek to find the time of maximum overlap for each blob, $t_c^i$, which we define as the time index in which the blob is

maximally overlapping with the Greenland Ice Sheet. The area of the GrIS covered by blob $i$ for time $t$ is,

$$a^i(t) = \sum_{n=1}^{ncol} \Delta a_n^i(t) \qquad (2)$$

where $\Delta a_n^i(t)$ is the overlap area between the GrIS and blob $i$ for each grid cell $n$,

$$\Delta a_n^i(t) = f_n \Delta A_n B_n^i(t) \qquad (3)$$

and $\Delta A_n$ is area of each grid cell and $f_n$ is the fraction of each grid cell covered by the GrIS. The time of maximum overlap $t_c^i$

is the time index $t$ for each blob $i$ where $a^i(t)$ is a maximum. Of course, not all blobs descend upon the GrIS throughout their

lifetimes. We therefore redefine $i$ to denote the subset of blobs that intersect the GrIS at some point during their lifetime.

To integrate any arbitrary horizontal variable $x_n(t)$ over the entire GrIS overlap area, coinciding with blob $i$ in the vicinity

of the time of maximum overlap $t_c^i + \delta t$,

$$X^i(t_c^i + \delta t) = \sum_{n=1}^{ncol} x_n(t_c^i + \delta t) \Delta a_n^i(t_c^i + \delta t), \qquad (4)$$

whereas the area average value of the variable $x_n$ for blob $i$ is,

$$\bar{X}^i(t_c^i + \delta t) = \frac{\sum_{n=1}^{ncol} x_n(t_c^i + \delta t) \Delta a_n^i(t_c^i + \delta t)}{\sum_{n=1}^{ncol} \Delta a_n^i(t_c^i + \delta t)}. \qquad (5)$$



The time of maximum overlap $t_c^i$ is used to provide a common reference time for averaging the integrated quantities $X^i$ over all blobs.

We ran this AR characterization process over each of the four ensemble members (ESMF-f19, ESMF-ne30pg2, TempestRemap-f19,
TempestRemap-ne30pg2) and took the average of each variable over the entire ensemble.

## 2.4 Validation

Reanalysis data from ERA5 and MERRA2 were used to validate the ensemble generated AR variables. The same remapping and compositing workflow that was applied to CESM2.2 simulations was applied to reanalyses. Meteorological reanalysis datasets combine observational data with a numerical atmosphere model to interpolate spatially and temporally onto a global
grid. ERA5 is the fifth reanalysis dataset produced by the European Centre for Medium-Range Weather Forecasts (Hersbach et al., 2020). ERA5 data has horizontal spatial resolution of roughly 27 km and the variables chosen for this study have hourly resolution, though we reprocessed this to six-hourly to match the timesteps in the CESM2.2 model outputs.

Similar to ERA5, the Modern-Era Retrospective Analysis for Research and Applications, version 2 (MERRA2) uses available satellite data, observational data, and the Goddard Earth Observing System (GEOS) model to provide users with a spatially and
temporally complete datset (Gelaro et al., 2017). MERRA2 has horizontal resolution of 56 km (latitude) × 69 km (longitude) and three-hourly temporal spacing, which we also reprocessed to six-hourly.

These two reanalysis datasets were chosen as validation due to their frequent application in prior studies (Bresson et al., 2022; Collow et al., 2022; Viceto et al., 2022; Zhou et al., 2022; Mattingly et al., 2023). The CESM2.2 model data and ERA5 share an overlapping study period of 1979-1998. Given that the available MERRA2 data begins in 1980, we chose to include
data available from 1980-1999 in order to maintain the same number of years in our study period (1979-1998).

## 3 Results

### 3.1 Frequency, Seasonality, and Origin Locations of Atmospheric Rivers

Between 7500 and 10100 ARs were detected in the Northern Hemisphere across the six model configurations and the two reanalysis products between the years 1979-1998 (1980-1999 for MERRA2) (Figure 3). As MERRA2 includes a different year
(1999) than the modeled outputs and ERA5, we ensured that this year experienced a number of ARs which did not vary greatly from 1979-1998 before including it in our analysis. MERRA2 resolved the highest number of ARs at 10094 and f19 the lowest at 7514. We used the number of ARs intersecting the GrIS (Table 2) and ARs detected globally to calculate the percentage of ARs intersecting the ice sheet. This metric only varied from 4.0% to 5.4%, with ERA5 showing the lowest percentage of ARs reaching GrIS.
The annual number of ARs intersecting the Greenland ice sheet ranged from 10-37 depending on grid-configuration and specific year. There are large variations from year to year among the grid configurations, as is expected. The reanlayses produce



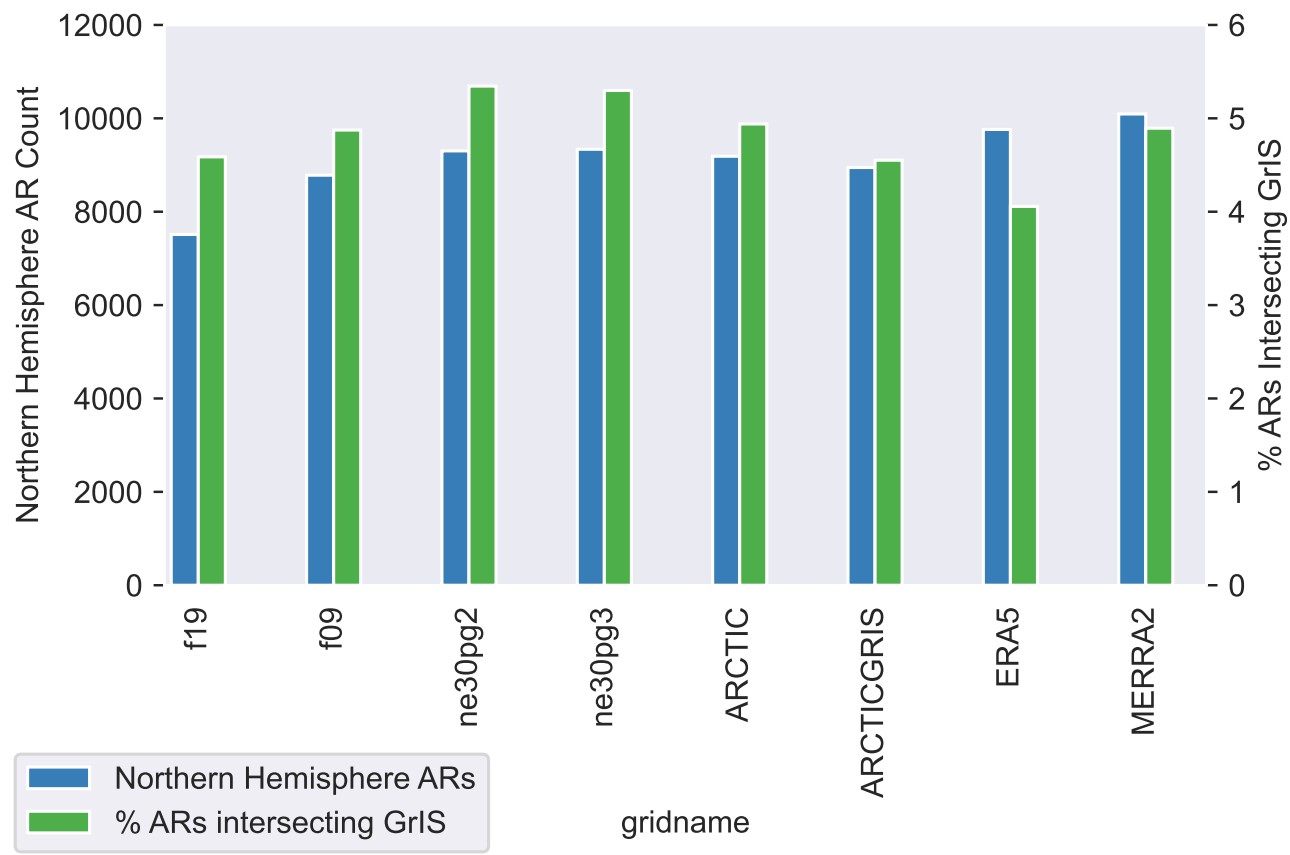

**Figure 3.** Average number of ARs in the Northern Hemisphere among the ensemble (left axis, blue), revealing a fairly consistent percentage of ARs traveling over the GrIS. Average percentage of ARs intersecting GrIS among ensemble (right axis, green) normalized by total ARs was calculated using data available in Table 2.

annual variations similar to the spread of modeled simulations, therefore suggesting that the models are producing ARs within or close to the bounds of reanalysis products.

The seasonal distribution of ARs reaching Greenland indicates that winter and spring generally have fewer ARs than summer
and fall (Figure 4). ERA5 produces the least ARs in all seasons except for winter. One or both VR grids produce the same median values as the reananlyses in every season. The QU unstructured grids produce the largest number of outliers of the grid configurations. In comparing the median values seasonally, Fall produces the most similar median number of ARs among the simulated outputs compared to the reanalyses.

Figure 5 shows the locations at which each AR that eventually intersects the GrIS formed during summer months. Most
ARs intersecting the GrIS during these months form over the central United States from around 30-45° latitude. The next most frequent location for AR formation is over the western Atlantic at similar latitudes. While ARs are defined to originate





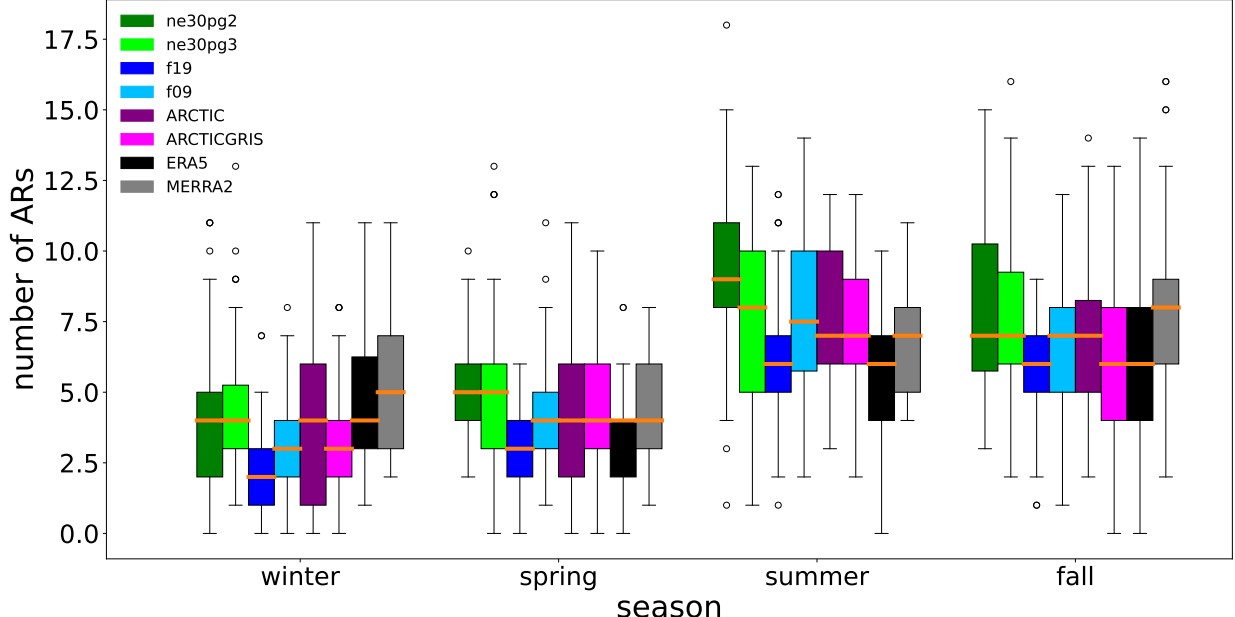

**Figure 4.** Number of ARs intersecting the Greenland ice sheet by season, with seasonal peaks in summer and fall. Winter was characterized as December through February, spring as March through May, summer as June through August, and fall as September through November. Seasonal distributions consider 20 years of data (1979-1998) using values from each of the four remapped ensemble members (N=80). Orange line in the center of each box signifies median value and box lower/upper boundaries describe the 25% and 75% quartiles, respectively. The whiskers extend from the box by 1.5x the inter-quartile range.

in mid-latitudes and transport water vapor poleward, the detection algorithm identifies a small number of airmasses with IVT characteristics above our detection threshold which originate at high latitudes. If these persist between timesteps the combination of the detection algorithm and the stitching algorithm designates them as ARs and they are retained in our analysis.

The reanalyses suggest more of the ARs that reach GrIS form over the equatorial Atlantic (Figure 5. Additionally, the differences among ensemble members in the reanalyses are much less evident due to the two remapping algorithms providing nearly identical answers, and causing the AR location points to largely overlap each other.

## 3.2    Areal Extent of Atmospheric Rivers

To study ARs intersecting the GrIS, we first compute the areal extent of each blob overlapping the GrIS. The coarser resolution

LL and VR grids have smaller footprints compared to their higher resolution pairs, while the QU simulations show the opposite relationship (Table 3). The VR simulations have the smallest footprints, hence are most similar to the reanalyses. In nearly all cases remapping to the QU ne30pg2 grid yields smaller footprints than remapping to f19 (Table 3).





**Figure 5.** Grid cell origin location for each JJA AR eventually intersecting the GrIS. Location dots vary based on color and size to signify number of ARs originating at that specific point and which ensemble member is represented, respectively. The smallest dots signify one AR formed in that grid cell and the largest signify ten ARs. Color and ensemble member pairings are as follows: dark blue- ESMF-f19, light blue- TempestRemap-f19, dark red- ESMF-ne30pg2, light red- TempestRemap-ne30pg2.





**Table 3.** Area of ARs intersecting GrIS

| grid name | f19 areal extent (km$^2$)$_a$ | ne30pg2 areal$_b$ extent (km$^2$) | average areal$_c$ extent ($10^5$ km$^2$) |
|---|---|---|---|
| f19 | $1.09 \times 10^6$ | $9.37 \times 10^5$ | 10.1 |
| f09 | $1.25 \times 10^6$ | $1.17 \times 10^6$ | 12.1 |
| ne30pg2 | $1.33 \times 10^6$ | $1.18 \times 10^6$ | 12.5 |
| ne30pg3 | $1.05 \times 10^6$ | $9.82 \times 10^5$ | 10.2 |
| ARCTIC | $8.55 \times 10^5$ | $8.67 \times 10^5$ | 8.6 |
| ARCTICGRIS | $9.80 \times 10^5$ | $8.46 \times 10^5$ | 9.1 |
| | | | |
| ERA5 | $6.07 \times 10^5$ | $5.11 \times 10^5$ | 5.6 |
| MERRA2 | $7.11 \times 10^5$ | $6.29 \times 10^5$ | 6.7 |

**Table 3.** [a]Values are the average of each of the f19 ensemble members (ESMF-f19, TempestRemap-f19).

[b]Values are the average of each of the ne30pg2 ensemble members (ESMF-ne30pg2, TempestRemap-ne30pg2)

[c]Values are the average of each of the four ensemble members (ESMF-f19, ESMF-ne30pg2, TempestRemap-f19, TempestRemap-ne30pg2)

The variation of footprint size is mainly due to the spatial distribution of ARs across the GrIS (Figure 6). ARs most frequently make landfall with the southwestern and southeastern margins of the GrIS, and the number of ARs per grid cell rapidly declines moving inland for all configurations. ARs modeled with LL and QU grid configurations travel further inland than in the VR grids and reanalyses. It should also be noted that fewer ARs make landfall in the northern portions of the GrIS in ERA5 than any of the other configurations. This lack of northern ARs explains why ERA5 has the lowest areal extent in Table 3.

### 3.3 Number and size of atmospheric rivers

In order to composite precipitation associated with ARs over the GrIS, we first determine the number and duration of storms overlapping land. Figure 7a describes the number of ARs that eventually intersect the GrIS based on days relative to time of maximum overlap, and Figure 7b shows the occurrence of these intersecting the GrIS relative to the time of maximum overlap. Five days before the time of maximum overlap roughly 20-25% of the landfalling ARs have formed (not shown). This number of ARs increases until the time of maximum overlap, with the largest increase from five days to two days before the time of maximum overlap (Figure 7a). This increase up to one day before the time of maximum overlap is likely due to ARs forming at high latitudes (Figure 5).

After the time of maximum overlap, the number of ARs decreases for all grid configurations and reanalyses. The number of ARs one day after the time of maximum overlap is 25-50% lower than the number of ARs during time of maximum overlap.





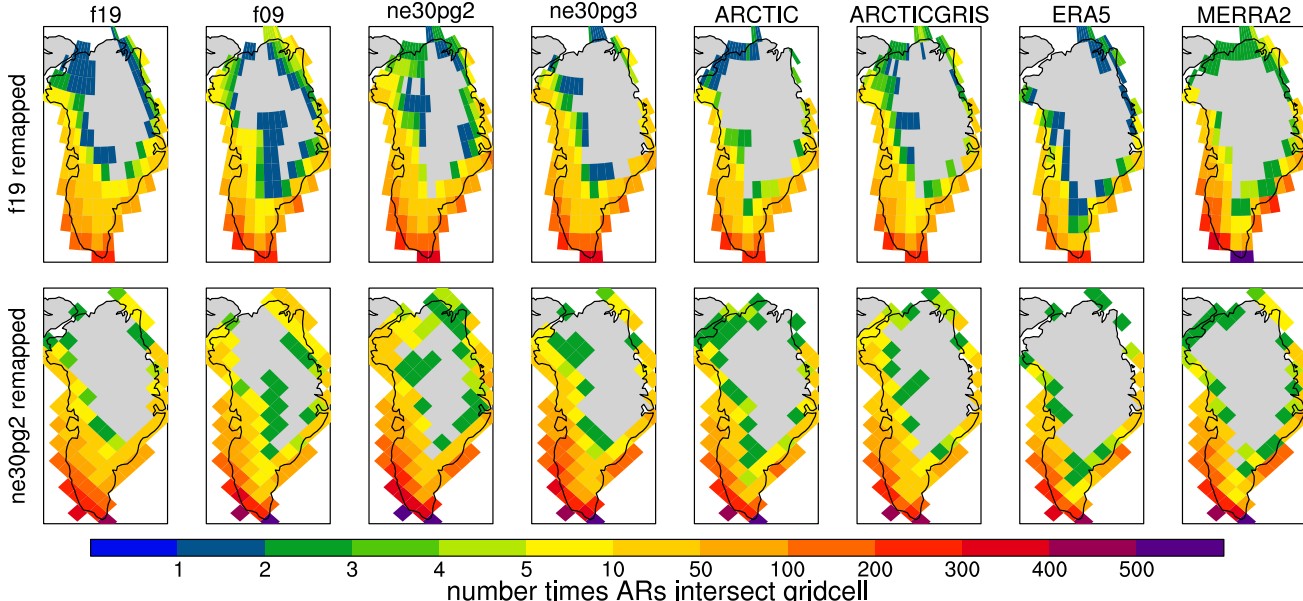

**Figure 6.** Spatial distribution of ARs over the GrIS using grid configurations remapped to f19 and ne30pg2. Most ARs make landfall in the southwest of Greenland.

This means that many of the ARs rapidly dissipate, indicating that a large amount of moisture is being transferred from the ARs to the GrIS, although some ARs do continue evolving until around five days past the time of maximum overlap.

Figure 7b illustrates the average area of the ARs relative to the time of maximum overlap on a log scale. Two days before maximum overlap there is a consistent and smooth increase in AR size for all grid configurations and the reanalyses (Figure 7b). This increase continues until one day before maximum overlap where all configurations produce a sharp decrease in AR size due to a rapid reduction of moisture. After the time of maximum overlap all the simulations and reanalyses produce ARs slowly increasing in size again. The QU configurations produce the largest ARs for almost the entire study period. The

reanalyses produce smaller AR areas, especially after the time of maximum overlap.

    The peak storm count at time of maximum overlap in Figure 7c is equal to the ensemble average of storm counts in Table 2. The QU grids produce more ARs than the rest, with the LL, VR, and MERRA2 in the middle, and ERA5 producing the least. Figure 7c also shows that the majority of ARs pass over Greenland in two days, supported by previous research (Box et al., 2023). However, it seems that outside of the +/- one day from maximum overlap, the agreement between outputs degrades.

Additionally, outside of that one day window few ARs are actually overlapping the GrIS (< 10 ARs). Thus, needing a larger sample size to calculate meaningful statistics later on, we chose to analyze the ARs over the course of a +/- one day window.

    The area of an AR overlapping with the GrIS also varies during its lifespan (Figure 7d). In general only a very small portion of any AR overlaps with the GrIS. Average AR areas range from 140-200x1$^{10}$ m$^2$ but less than 5.0x1$^{10}$ m$^2$ of any AR is overlapping with the GrIS even during its time of maximum overlap. The f19 simulations have the largest area of overlap





during the time of maximum overlap and onward despite it not having the largest AR area (Figure 7c). Despite the QU grids producing the largest ARs (Figure 7c), they do not have the largest overlap area with the GrIS. Reanalyses and the VR grids consistently produce smaller overlap areas.

## 3.4 Precipitation

Many ARs affecting Greenland make landfall on the west coast and travel eastward until they reach the steepest portion of
the GrIS (Figure 8). At this point, much of the moisture deposits as precipitation and the storm dissipates. Figure 8 shows the composite precipitation map (from CAM) of all ARs as they travel over their storm path for one particular grid configuration and remapping scenario. The main difference between configurations comes from how far inland ARs can penetrate.

Figure 9 shows examples of 95th percentile ARs intersecting the GrIS, as well as the precipitation rates and sea level pressures during time of intersection for all grid configurations. We see the behavior of ARs intersecting on the SW, S, and SE
of the GrIS. Additionally, we see the increased pressure gradients associated with many of these ARs. Many of the snapshots of ARs captured in figure 9 show the impact of ARs intersecting the GrIS on precipitation, as the AR is only grazing the GrIS at a point in time yet precipitation can be seen throughout a larger portion.

Figure 10a shows the cumulative AR precipitation expressed as the average water equivalent depth per grid cell from CAM, with respect to the time of maximum overlap (equation 5). Precipitation depth steadily increases over the one day study period
(Figure 10). After the study period, there is a difference of around 30 mm between the highest and lowest depths produced by mean values from the grid configurations and reanalyses. The configuration f09 produces the highest depth of precipitation while MERRA2 and f19 produces the lowest. ERA5 also produce magnitudes and trends of depth accumulation similar to the six modeled outputs.

Figure 10b compares the cumulative AR precipitation expressed as the 95th percentile water equivalent depth per grid cell.
Precipitation depth increases over the study period in a similar way as figure 10a, but on a different scale due to figure10b being the extreme ARs. At the end of the study period, the lowest and highest extreme ARs differ by about 40 mm, which is similar to the mean ARs. Aside from the scales, the main difference between the mean and extreme depths is the ordering of the outputs. ARCTICGRIS, ARCTIC, and f09 produce larger depths than MERRA2 and ERA5. This could possibly be related to the model outputs being calculated using 6-hourly instantaneous whereas the observation-based data uses 6-hourly averages.

Figure 10c compares the average area-integrated cumulative precipitation (equation 4), showing variation among model outputs and the two reanalyses. Area-integrated precipitation varies from around 0.7 Gt in ERA5 to 2.5 Gt in f19. The two QU grids produce precipitation on the higher end of the spread followed by the other LL grid, f09. The two VR grids simulate lower area-integrated cumulative precipitation than the other model grids. Both reanalyses produce less precipitation compared to the CESM2.2 model grids, though MERRA2 produces similar precipitation magnitudes to ARCTICGRIS. There is a difference
of about 0.1 Gt between ARCTICGRIS and MERRA2 and about 0.4 Gt for ARCTICGRIS and ERA5. The trends in rate of increase of area-integrated precipitation are different than those seen in the average precipitation Figure 10a; the highest rate of increase is during the day preceding maximum overlap for all grid configurations except for f19, after which it begins to slow.





**Figure 7.** (a) Number of ARs that eventually intersect GrIS, i.e., one day following the time of maximum overlap, roughly 40% of ARs that intersected GrIS dissipate and (c) days that ARs are overlapping GrIS, showing rapid dissipation after landfall and an average overlap with GrIS of about 2 days. Total number of ARs intersecting GrIS at time t = 0 is equal to average number of intersecting ARs for each grid configuration in Table 2. (b) Area (km$^2$) of ARs which eventually intersect GrIS on logarithmic scale and (d) area (km$^2$) of ARs which overlap the GrIS during landfalling, showing that only a small portion of the AR overlaps the GrIS.





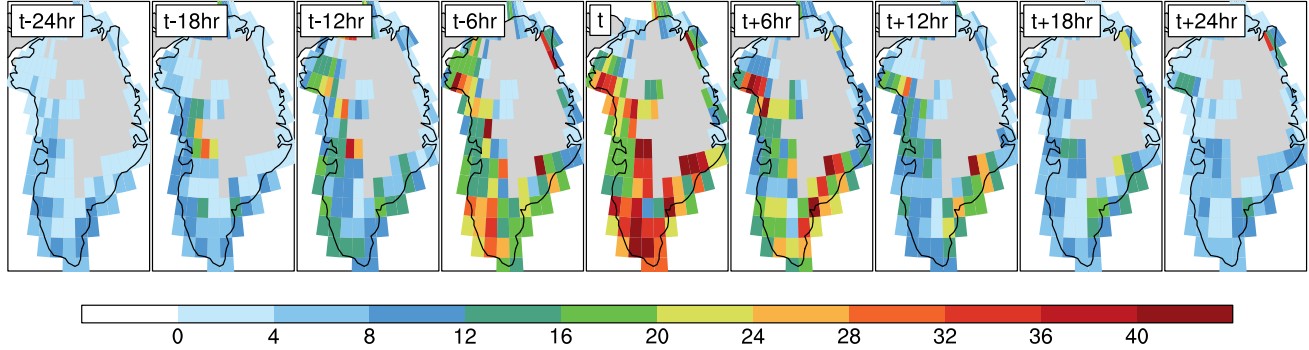

**Figure 8.** CAM precipitation rate over the GrIS during landfalling ARs, providing an example from the ARCTICGRIS grid of how far the precipitation from ARs travels inland. Rate considers each landfalling AR and finds average of all storms. Cumulative precipitation is integrated over area and time. In the case of this configuration (ARCTICGRIS mapped to f19 using ESMF), 520 ARs made landfall with the GrIS; this figure shows the average precipitation rate of all 520 ARs. Time t indicates the point at which the AR is maximally overlapping the GrIS and time is projected into the past and future.

Figure 10d compares the 95th percentile extreme area-integrated cumulative precipitation. Similarly to figure 10a and 10b, the precipitation magnitudes between the mean ARs and extreme ARs are also very similar. ARCTICGRIS and MERRA2 are the most similar model outputs to MERRA2 and ERA5. In particular, ARCTICGRIS and MERRA2 only differ by around 0.5 Gt in the extreme ARs.

Figure 11a shows the 2 day area-averaged cumulative precipitation as the average water equivalent depth per grid cell from CAM for average sized ARs with respect to the radial great circle distance of GrIS grid points to the AR in km. In particular, for each AR at time t, we include all GrIS grid cells that are within a given great circle distance to any point within the AR, in compositing the precipitation. We see that the area-averaged precipitation does not greatly decrease within 100 km of the AR. From around 200 km to 500 km, the precipitation steadily decreases. From 500 km onward, the precipitation seems to decrease at a slower rate. We see that all model outputs and reanalyses exhibit similar behavior, mainly differing in maximum area average-precipitation, with f09 have the largest and MERRA2 the smallest.

Figure 11c shows the 2 area-integrated cumulative precipitation as the average water equivalent depth per grid cell from CAM for average sized ARs with respect to the radial great circle distance of GrIS grid points to the AR in km. Similarly to the area-averaged precipitation, this integrated precipitation does not increase from 0 km to 100 km. From 200 km to 500 km, the precipitation increases at a higher rate than is seen in this viewpoint of 0 km to 1200 km. In combining figure 11a and 11c, we can estimate that most precipitation which is associated with an AR occurs within 500 km of that storm. At this 500 km mark, the reanalyses produce around 4 Gt of precipitation and both VR outputs replicate this magnitude well. The LL and QU produce roughly 4.4 to 5.8 Gt at 500 km and the differences between the VR and LL/QU are even larger at the 1200 km distance.



**Figure 9.** 95th percentile ARs and precipitation rates produced by LL, QU, and VR configurations at four different datetimes. ARs are outlined in blue. Black contours are sea level pressure anomalies with 5 hPa intervals. Datetimes are given in YYYYMMDD, where Y is year, M is month, and D is day.



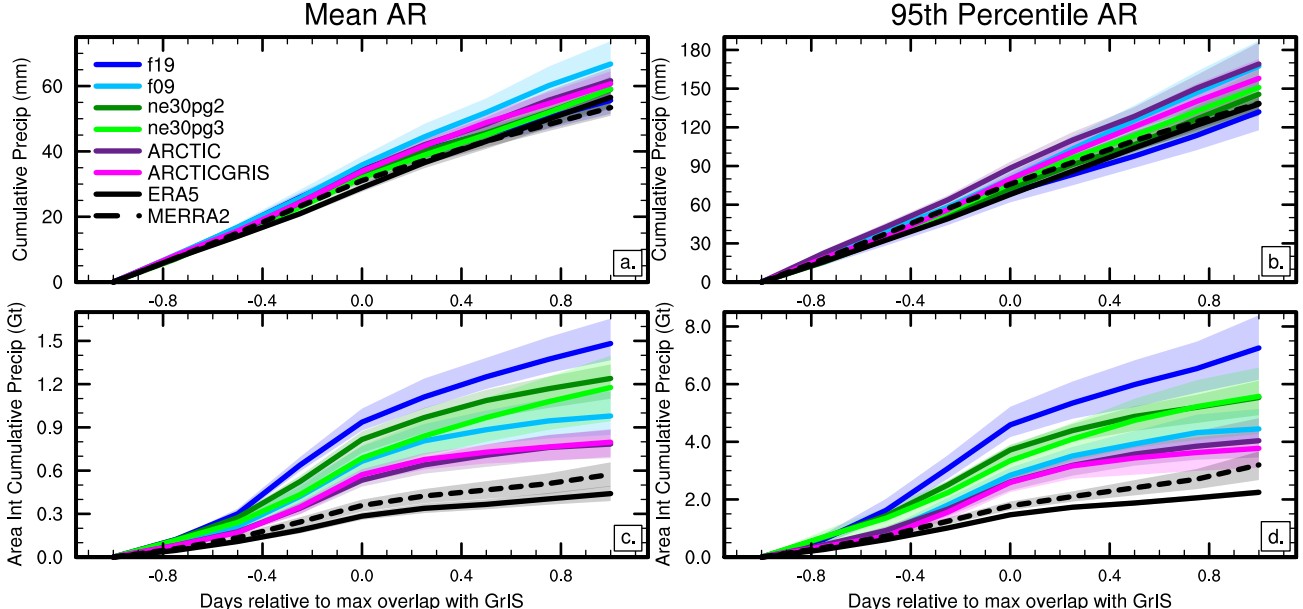

**Figure 10.** (a) Mean area-average precipitation and (c) mean area-integrated cumulative precipitation over GrIS during landfalling ARs, displaying a small spread in spatially averaged precipitation among the grids but a larger spread in cumulative precipitation given the differences in AR size. (b) 95th percentile area-average precipitation and (d) 95th percentile area-integrated cumulative precipitation of GrIS. Area-average precipitation considers each landfalling AR and finds average (a) and 95th percentile (b). Cumulative precipitation integrates over area and time and finds average (c) and 95th percentile (d). Time t indicates the point at which the AR is maximally overlapping the GrIS. Precipitation is derived from six-hourly instantaneous samples from the variable PRECT for ERA5, PRECTOT for MERRA2, PRECC + PRECL for all modeled simulations.

Figure 11b shows the area-averaged precipitation vs radial great circle distance for the 95th percentile ARs. the behavior is similar to the average sized ARs in that there is a breaking point in the slope around 500 km, indicating that most AR-precipitation occurs within this distance for extreme ARs as well. We see that from 0 km to 200 km there slope for both reanalyses is flat.
This may be due to the 6-hourly average sampling instead of instantaneous, where we estimate the effects on the VR ARCTIC in the dotted purple line in Figure 11b, which was computed using 2 point running average in time.

Figure 11d shows the area-integrated precipitation vs radial great circle distance for the 95th percentile ARs. Aside from the large magnitudes with the extreme ARs, this behavior is extremely similar to average ARs. We see that at the 500 km mark, the reanalyses produce roughly 13 Gt precipitation, which is extremely well captured with VR outputs. At 500 km, the LL and
QU grids produce between 15-17 Gt precipitation.



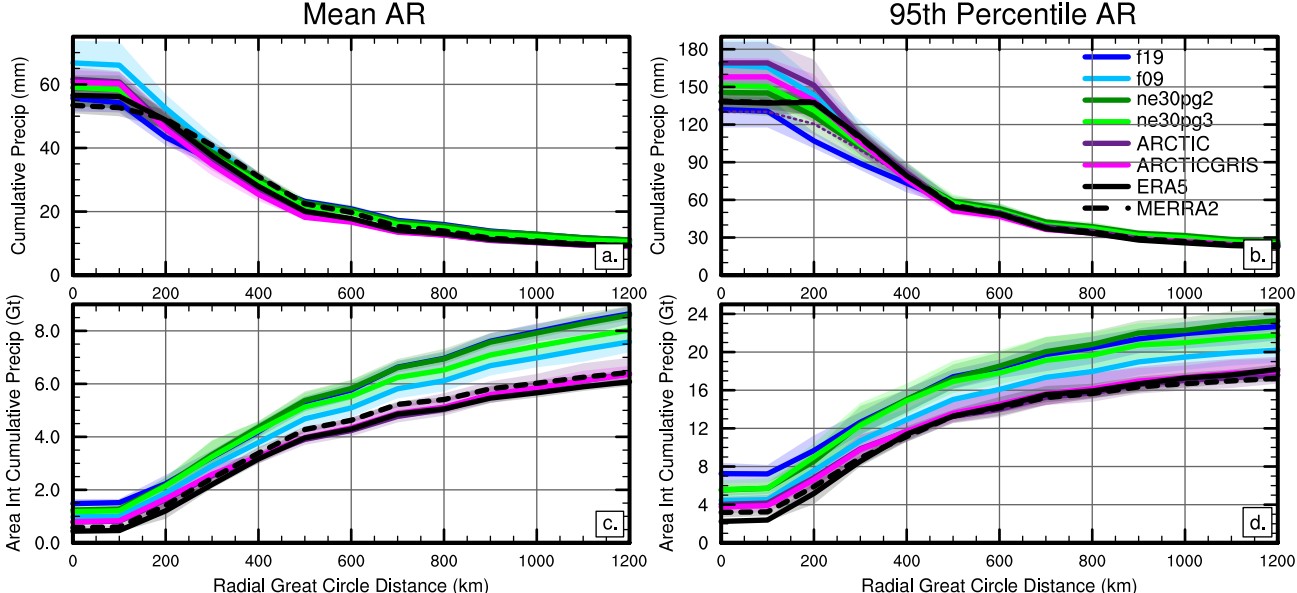

**Figure 11.** (a) Mean area average precipitation and (c) mean area-integrated cumulative precipitation over GrIS compared to radial great circle distance of GrIS grid points to AR, displaying large amounts of precipitation occurring within 500 km of AR that can be attributed to that storm. (b) 95th percentile area-average precipitation and (d) 95th percentile area-integrated cumulative precipitation over GrIS compared to radial great circle distance of GrIS grid points to AR, showing similar findings of mean AR with precipitation 500 km away from AR being attributed to that storm. Area-average precipitation considers each landfalling AR and finds average (a) and 95th percentile (b). Cumulative precipitation integrates over area and time and finds average (c) and 95th percentile (d). Radial Great Circle Distance (km) describes the distance of each grid point on GrIS to AR. Precipitation is derived from six-hourly instantaneous output in the model runs, whereas the reanalyses uses 6-hourly averaged variables. The dotted purple line in (b.) is the ARCTIC run but using using two-point averaging to estimate the impact of using averaged variables in the reanalyses.

## 4 Discussion

The difference in topographic resolution among grid configurations likely explains the variation in AR areal extents over Greenland (Figure 2). Coarser grids require more topographic smoothing to prevent the excitation of error-prone grid scale modes in the dynamical core (Lauritzen et al. 2015). In the LL and QU grids, topographic smoothing is ubiquitous across the

GrIS and allows for ARs to penetrate further into the interior of the ice sheet, reducing orographic lifting that would otherwise drain the ARs of their moisture and cause them to dissipate (Box et al., 2023). For example, the LL f19 grid has the lowest maximum elevation for the GrIS and the largest AR areal extent. In contrast, the VR grids and reanalysis datasets all have similar topography, capturing high elevations abd steep elevational gradients across the GrIS.

We suggest that the higher and steeper topography resolved in VRs and the reanalyses prevent ARS from penetrating as far

inland as the LL and QU grids. The finer resolution VR grids and reanalyses produce smaller ARs (Figure 3c), consistent with



more precise tracking of atmospheric moisture. Furthermore, the large GrIS overlap of ARs in f19 (Figure 3d) is not related to the size of ARs prior to landfall (Figure 3c), supporting the hypothesis that topographic smoothly explains the variations in AR areal extent. Previous studies echo this idea, finding that VR grids better resolve climate and snowpack in regions of complex topography in California (Huang et al., 2016; Rhoades et al., 2020b).

The differences in area-integrated cumulative precipitation among grid configurations (Figure 10) reflect the areal extents of ARs over the GrIS (Table 3, Figure 6). As the average precipitation depth among grid configurations is similar among all grids, simulations which cover a larger areal extent of the GrIS deposit more total precipitation. ERA5 produces the lowest integrated precipitation, followed by MERRA2 and both VR grids, with the LL and QU grids producing the most precipitation. The CESM2.2 simulations all overestimate precipitation compared to reanalyses, in line with previous findings (Herrington 315     et al., 2022; van Kampenhout et al., 2020). We also find that not all precipitation related to ARs occurs directly below the AR. Figure 11 shows that precipitation from ARs likely occur within 500 km of the AR detected by our methods. This note is useful for informing future methodology in detecting precipitation from ARs.

Additionally, extreme AR precipitation tends to follow similar trends as for average precipitation, as both are driven by the areal overlap. The most notable difference comes from the extreme precipitation depths for the VR grids being larger and 320     less closely matching the observation based data, though this could be due to the different sampling periods between the two outputs. Keeping these facts in mind, future studies of land-falling ARs will benefit from either high resolution configurations such as VR grids or bias correction for precipitation when calculating mean AR precipitation.

In this study, many ARs eventually intersecting the GrIS initially form over the mid-latitude central United States (Figure 5), consistent with Neff et al. (2014). The TempestExtremes tracking algorithm also identified a subset of ARs at uncharacteristically 325     high latitudes, suggesting that a more polar-optimized tracking algorithm should be used around Greenland (Shields et al., 2023). Alternatively, this high-latitude AR detection might challenge the typical definition of ARs- does an AR need to format mid-latitudes? Or are there polar specific variants to consider, as Komatsu et al. (2018) suggests? Path-tracking would be a useful tool to study AR formation and movement patterns to better understand how they will impact GrIS SMB.

Our analysis shows ARs overlapping the GrIS for around 1.5 to 2 days, with most overlapping in the +/- 1 one window. A 330     study from Zhou et al. (2018) found that the mean lifespan of long-traveling ARs in the North Pacific is at least 3 days (72 hours). As most ARs intersecting Greenland travel for multiple days before reaching the GrIS in our simulations, our study aligns with these findings. Mattingly et al. (2020) and Box et al. (2023) also found that ARs generally persist over Greenland for 1-2 days, though their impacts persist for multiple days after the AR dissipates or moves away from Greenland. The August 2021 AR over the GrIS is an example of a two-day overlapping AR with strong glacio-hydrological impacts that endured the 335     rest of that melt season (Box et al., 2022). We suggest that to study ARs over the GrIS in a statistically significant way, the +/- one day overlap window should be used.

ERA5 and MERRA2 differ in geographic distribution of ARs over the GrIS, suggesting the need to consider multiple reanalyses when studying precipitation from ARs in Greenland. While VR grids and MERRA2 produce many ARs making landfall in the northern regions of the GrIS, ERA5 shows very few. These geographic landfall variations likely explain the 340     difference in areal extent between the VR grids and ERA5. Loeb et al. (2022) compared reanalysis products including ERA5





and MERRA2 and found that ERA5 had higher correlation to observational data, especially in southwestern Greenland. This suggests that to most accurately study ARs and precipitation in Greenland, multiple reanalyses should be used in combination with each other depending on regional performance.

The CESM2.2 simulations consistently produced more ARs and precipitation than ERA5. Extending the domain further southward to include more of the Atlantic Ocean might help close this gap. Rhoades et al. (2020a) found that increasing the domain of a VR grid focusing on the western US to include more of the Pacific Ocean lowered IVT and precipitation. Stansfield et al. (2020) noted similar effects of VR refined grid domain size on precipitation in the eastern US. However, Rhoades et al. (2020a) also found that despite these differences in IVT, AR characteristics and impacts on snowpack remained similar among the simulations. More investigation of VR grid extent is needed to understand how the grid domain affects ARs and Greenland surface mass balance.

## 5 Conclusions

This study uses CESM2.2 simulations from Herrington et al. (2022) to compare six grids in modeling ARs and related precipitation over the GrIS. The 1–2° LL grids configurations provide enhanced resolution over polar regions, though a polar filter is used to prevent numerical instability thus reducing some of this increased resolution. Two QU unstructured grids maintain roughly 1–2° uniform resolution throughout the globe. To study the impact of resolution on ARs around the GrIS, we compare simulations using these four coarser grids to two VR grids using the SE dycore, ARCTIC and ARCTICGRIS. Both VRs use 1/4° spacing over the Arctic and ARCTICGGRIS uses 1/8° grid over Greenland.

We developed a method which maps all output to the two coarsest model grids, and using two different remapping methods to account for uncertainty of comparing AR statistics in model simulations and reanalysis products across vastly different grids. We use the overlap area of an AR and the GrIS to determine how AR characteristics and precipitation varies based on grid configuration. This method attributes precipitation from regions of the GrIS which an AR is directly overlapping at a point in time and then takes the sum of all precipitation in each of these regions by grid configuration. This allows for a robust comparison of precipitation across grids with realistic uncertainty. This method can and should be applied to other variables relevant to ARs and the GrIS, including snowmelt and radiative fluxes (?Kirbus et al., 2023).

We find that the topography resolution of the grid likely constrains AR penetration into the GrIS. In coarser resolution grids, there is greater topographic smoothing of the GrIS and ARs can travel further inland. As the precipitation per grid cell does not vary greatly across grid configurations, the overlap extent of ARs largely determines the simulated precipitation falling onto the GrIS. Additionally, we see consistent patterns characterizing AR behavior and lifespan around the GrIS. In the CESM2.2 simulations and reanalyses, most ARs only intersect the GrIS for 1.5 to 2 days. ARs increase in intensity prior to landfall. One day before the time of maximum overlap, ARs experience a "draining period" and decrease in size, likely due to orographic uplift that drains the ARs of their moisture.

Finally, we find that the VR grids produce AR areal extents, cumulative integrated precipitation, and AR area that are most similar to the reanalysis datasets ERA5 and MERRA2. All CESM2.2 simulations produce higher values for all three AR



metrics than the reanalyses. Although VR grids deviate some from the reanalyses, VR grids outperform the LL and QU grids
used in our study and have resolutions approaching regional climate models but at lower computational costs. We therefore that
recommend modeling studies of ARs around Greenland consider using CESM2.2 VR grid configurations over uniform grids.

*Code and data availability.* The code and data presented in main part of this manuscript are available at https://github.com/adamrher/greenland-storms.

*Author contributions.* AW wrote manuscript and assisted with code preparation and ran analysis code. AH prepared methodology and
developed data processing code. EB secured project funding and resources for initial project conceptualization. All co-authors provided
edits and revisions to the manuscript, data analysis, and synthesis.

*Competing interests.* The authors declare that they have no conflict of interest.

*Acknowledgements.* Funding support for this work was provided in part by the National Science Foundation (1832959 and 2125868),
University of New Hampshire (UNH) Institute for the Study of Earth, Oceans, and Space, and UNH Space Grant.
This material is based upon work supported by the National Center for Atmospheric Research (NCAR), which is a major facility
sponsored by the NSF under Cooperative Agreement 1852977. Computing and data storage resources, including the Cheyenne supercomputer
(Computational and Information Systems Laboratory, 2017), were provided by the Computational and Information Systems Laboratory
(CISL) at NCAR.



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
