# Peer review of "Using variable-resolution grids to model precipitation from atmospheric rivers around the Greenland ice sheet"

_EGUsphere, 2023_

## Author Response (AR1)

Dear Dr. Wernli,

We graciously thank the reviewers for their feedback and ideas on this manuscript. We have incorporated the recommended changes and believe they have greatly improved the manuscript. In the attached pdf, changes have been tracked in  font and newly revised text in blue font. We have also included a clean, revised manuscript with markups.

Reviewer comments are shown in **bold**.
Author comments are in plain text.

Sincerely,
A Waling and Coauthors

Reviewer 1:

**This study evaluates atmospheric rivers (ARs) in simulations with varying resolutions. In its current form, I find the paper to be very technical and difficult to follow, so a significant revision of the text is required in my opinion. One of the main results is also not surprising; the authors state that "We suggest that the higher and steeper topography resolved in VRs and the reanalyses prevent ARS from penetrating as far inland as the LL and QU grids". This result is well known, as higher topography acts as a barrier to water vapor transport (IVT) and hence the inland penetration of AR features.**

Regarding our manuscript's technical nature, we have reduced technical language throughout all of the section and significantly revised the Section 2.1 - Model Simulations in the methods section (see Lines 85-114). Specifically we have removed more technical descriptions of CESM2.2 and have eliminated unnecessary abbreviations used in this section.

Additionally, we have added more information to the Abstract and Introduction that describe the importance of our work to the broader modeling community and de-emphasized the finding of one of the main results on higher and steeper topography. Revisions include:

> Lines 3-6: "In contrast with traditional modeling studies that rely on coarse (1° to 2°) uniform-resolution grids, this project investigates the effectiveness of variable-resolution (VR) grids in modeling ARs and their subsequent precipitation using refined grid spacing (0.25° and 0.125°) around the GrIS and 1° grid spacing for the rest of the globe in a coupled land-atmosphere model simulation."

> Lines 59-68: "This work will help the atmospheric community determine when the more computationally expensive (relative to coarse uniform grid spacing) but finer spatial resolution VR grids are most useful, especially important given the limited in-situ observations available for quantifying the effects of atmospheric rivers over Greenland on precipitation and surface mass balance. Models like RACMO2 (Noël et al., 2018) and other limited area models also provide high spatial resolution, but may be limited by imperfections in boundary conditions and limited in their ability to simulate climate feedbacks over longer multi-decadal time scales. In contrast, variable resolution grids provide an intermediate solution between coarse resolution coupled land-atmosphere models, such as CESM2.2, and fine-scale regional climate models using observation-based forcing data. The paper also details a replicable method for tracking ARs in the Atlantic Arctic region over a multi-decadal simulation, providing insight and guidance into the objective detection of ARs from model data."

Finally, we have done a careful, in-depth revision that aims to clarify any technical elements that we chose to still include in our manuscript. For example, we found our two categories of precipitation discussed throughout the text (area-averaged cumulative precipitation and area-integrated cumulative precipitation) to be wordy and confusing at times. Thus, we shortened "area-averaged cumulative precipitation" to "precipitation rate" (Line 296) and "average area-integrated cumulative precipitation" to "area-integrated precipitation" (Line 310). and use this terminology for clarity and easier reading.

Regarding one of our main findings (higher and steeper topography in finer resolution models) not being surprising, we have included additional studies that have reached this conclusion:

Line 73-75: "We hypothesize that the VR grids will simulate ARs more accurately than the coarser resolution grids through better resolution of fine-scale physical processes and topography, as has been seen in other studies investigating moisture intrusions in the Arctic (Ettema et al., 2009; Noël et al., 2018; Bresson et al., 2022).

Our study focuses on how well the models are capturing physical behaviors such as precipitation associated with orographic uplift. While there is extensive observation and modeling of ARs over the Pacific and California coast, the focus on ARs reaching Greenland is relatively new. Though one would assume orographic uplift would be the main factor influencing this AR derived precipitation in Greenland, our study shows how conventional coarse global grid configurations frequently used in the modeling community perform compared to Variable Resolution (VR) grids. We hope to persuade future modeling studies to consider using VR grids as they have been shown in our study to better resolve the well-known dynamics that you described.

Lines 25-28: "While there is extensive observation and modeling of ARs over the Pacific and California coast (Huang et al., 2016, 2020; Rhoades et al., 2020b), only more recently have studies focused on ARs reaching Greenland (Mattingly et al., 2018, 2020; Box et al., 2022, 2023; Kirbus et al., 2023; Mattingly et al., 2023)."

**I did not read the Editor's comments until after my review. I agree strongly with the Editor of the need for more physical interpretation of the results found, and possibly of case studies, especially of IVT and precipitation fields, to help the reader to follow your work.**

Thank you for your comment. We have significantly revised  the text associated with the 95th percentile cumulative precipitation (Figure 9), maps showing the 95th percentile ARs and their related pressure systems (Figure 10), and precipitation occurring outside the detected AR features (Figure 11) to include more physical interpretation of the results. We have summarized these changes below:

Lines 317-323: "A shortcoming of our approach is that we only composite the precipitation inside the tracked feature, however precipitation associated with an AR may include regions outside the tracked feature. Figure 10 shows snapshots from the models of the 95th percentile ARs near the time of their maximum overlap with Greenland, and the outline of the detected feature provided in magenta. The detected feature represents the moist core of the AR, which, unlike the larger synoptic system, does not overlap with a large portion of land at any point throughout its lifecycle (Figure 7d). The snapshots indicate the warm front out ahead of the AR core contributes a substantial amount of the storm's precipitation, which have been neglected from our precipitation composites thus far."

Line 324-331: "Figure 12a quantifies the impact of including regions outside the core of the AR in compositing precipitation due to that AR. It shows the precipitation rates over the two-day window with respect to the radius of the expanded composite area. If a GrIS grid point lies within a radial great circle distance to any point in the detected feature, it is included in the composite. From around 200 km to 500 km, the precipitation rates steadily decrease, as it incorporates regions with smaller magnitude precipitation rates in the composite. From 500 km onward, the precipitation rates seems to decreases at a slower rate, suggesting a transition to the marginal outer regions of the synoptic system which may not be exclusively associated with the storm itself. We see that All model outputs and reanalyses exhibit similar behavior, mainly differing in maximum precipitation rates, with LL_1∘ having the largest and MERRA2 the smallest."

We have also clarified why a climatological approach is necessary over historical case studies, given that CESM is not forced by reanalysis and thus cannot be directly compared to historical case studies:

> Lines 77-79: "In our study, the model output is compared to the climatology of ARs detected by ERA5 and MERRA2, two observation-based meteorological reanalysis datasets, as in other studies involving simulated ARs (Bresson et al., 2022; Viceto et al., 2022; Zhou et al., 2022; Mattingly et al., 2023)."

> Line 203-206: "It is important to emphasize that CESM2.2 simulations are free-running, coupled land-atmosphere climate simulations constrained by monthly sea-surface temperature and sea-ice extent, but not by meteorological observations or reanalysis. We therefore present climatological comparisons among model configurations rather than historical observation-based case studies."

**Please find some comments below which I hope are useful to you.**

**Line 20: "to the poles". I would suggest rephrasing to either "polewards" or "across the mid-latitudes".**

We appreciate this feedback and have implemented it.

**Lines 20-25: There may be further information for the Introduction in the European State of the Climate Report (https://climate.copernicus.eu/esotc/2022/greenland-heatwaves).**

Thank you for this reference, we have included it and additional information about the September 2022 atmospheric river in our introduction:

> Lines 38-40: "The GrIS experienced multiple major melt events in recent years, including one in August 2021 that was associated with rainfall at Summit Station (Box et al., 2022) and one in September 2022 when at least 23% of the GrIS experienced surface melt (C3S, 2023)"

**Introduction last 2 paragraphs: These do not set up the paper as well as could be expected. These paragraphs could be combined and revised to state clearly the aims or questions addressed in the study. I felt the first line of penultimate paragraph was past research, rather than this study.**

We have revised and consolidated the last two paragraphs of the introduction. We hope this provides a better set up for the paper's aims and purpose:

> Lines 69-82: "This study takes advantage of pre-existing model output from multi-decadal simulations and compares AR characteristics and precipitation produced by six grid configurations using the Community Earth System Model version 2.2 (CESM2.2) (Danabasoglu et al., 2020; Herrington et al., 2022): two latitude-longitude grids, two quasi-uniform unstructured grids, and two VR grids (Zarzycki and Jablonowski, 2015; Zarzycki et al., 2015). The VR grids used in CESM2.2 employ grid refinement to yield enhanced resolution around our region of interest, Greenland. We hypothesize that the VR grids will simulate ARs more accurately than the coarser resolution grids through better resolution of fine-scale physical processes and topography, as has been seen in other studies investigating moisture intrusions in the Arctic

(Ettema et al., 2009; Noël et al., 2018; Bresson et al., 2022). Accurately modeling precipitation from ARs is important because it has been suggested that during early summer nearly 40 percent of precipitation in Greenland is due to ARs (Lauer et al., 2023). In our study, the model output is compared to the climatology of ARs detected by ERA5 and MERRA2, two observation-based meteorological reanalysis datasets, as in other studies involving simulated ARs (Bresson et al., 2022; Viceto et al., 2022; Zhou et al., 2022; Mattingly et al., 2023). Section 2 describes the model grids, remapping workflow, AR detection method, precipitation counting method, and the validation datasets used in this study. Section 3 contains the main results and analyses performed in this project. Section 4 discusses the implications of these results. Section 5 summarizes main conclusions from our work and provides direction for future research."

**Methods: CESM2.2 model output used. Then you say "atmosphere simulations used the Community Atmosphere Model 6.3 (CAM6) (Craig et al., 2021)". Please clarify. The methods are not clear on the models. Please revise.**

We have revised Section 2.1 to more clearly indicate that CLM5 and CAM6 are the land surface and atmosphere model components, respectively, coupled within CESM2.2 Earth System Model. The revised text is highlighted in blue in the resubmitted manuscript (see Lines 92-99):

**Lines 92-95: Which simulations? CESM2.2 or CAM6? Are they four ensemble members for each configuration in Figure 2? What are the acronyms for the simulations? Are these bilinear, conservative, or some other remapping technique?**

We have clarified that CAM6 and CLM5 are, respectively, the atmospheric model and land surface model components of CESM2.2. The topography shown in Figure 2.2 is generated for CLM5 and used as a boundary condition for CAM6, which is coupled to CLM5 in the CESM2.2 model environment. The text in Figure 2 has been adjusted to ensure that readers understand that the topography shown is native to each CESM2.2 grid configuration and therefore does not undergo any remapping to get these visualizations. The acronyms for each grid have also been updated in Figure 2 and throughout the manuscript, as suggested by Reviewer #2. We have also clarified that the native topography is used as a boundary condition for CAM6 (Lines 107-108). Please see revised Figure 2 and associated caption below:

[Figure]

**Figure 2.** Native topography of each CESM2.2 grid configuration and reanalysis dataset used in this study, with higher resolution grids more accurately capturing the elevation gradients in Greenland. A-b show latitude-longitude (LL) (a- LL_2°, b- LL_1°) grids, c-d quasi-uniform (QU) (c- QU_1.5°, d- QU_1°), and e-f variable-resolution (VR) (e- VR_0.25°, f- VR_0.125°), g shows ERA5, and h shows MERRA2.

Remapping techniques are both conservative; text has been revised to make this more clear:

> Line 115-120: "To control for the sensitivity of the atmospheric feature detection algorithm to grid structure and resolution, we remapped the output from each simulation to the coarsest LL grid (LL_2°) and the coarsest QU grid (QU_1.5°) using two remapping methods. This was a cautious choice as mapping to higher-resolution grids is inaccurate for first-order methods. The two remapping methods were ESMF (Team et al., 2021) and TempestRemap (Ullrich and Taylor, 2015), both of which use conservative formulations."

**Line 133: CLM. Is this CLM5?**

Yes, CLM5. This has been specified.

**Figure 4: The whiskers would be better as a percentile (e.g. 1 and 99 percentiles, or 5 and 95 percentiles). The reader will find it difficult to interpret 1.5 x interquartile range.**

We appreciate this feedback but feel that the IQR is an appropriate way to display this data. We find that by using the IQR we get a more accurate view of the 25th to 75th quartile AR seasonalities and are less affected by outliers.

**Line 185: It does not look like ERA5 produces the fewest ARs in Fall or possibly Spring. Do you mean in the average number which is not given in the boxplots?**

We have revised this section of the manuscript to more accurately reflect the seasonal distribution of ARs presented in Figure 4:

> Line 219-220: "The seasonal distribution of ARs reaching Greenland indicates that winter and spring generally have fewer ARs than summer and fall (Figure 4). One or both VR grids produce the same median values as the reanalyses in every season."

**Figure 5: How are these source points calculated? This needs further explanation, especially on what the physical processes would be to have AR sources in these regions. Are there references for these processes (e.g., Neff et al 2014 is given later) or regions? There are certain simulations with points in Mexico – is this unrealistic? Does the difference between the reanalyses and the other simulations suggest a bias in the models which is not realistic? Please provide further explanation.**

Source points are calculated by finding the grid point with the largest IVT contained within the tracked feature, at the first time the feature is detected. This has been added to the text accompanying Figure 5 on page 12:

> "The origin locations are detected by searching for the grid cell with the maximum IVT inside the AR at the first time that the AR is detected. Note that the location at which an AR forms is sensitive to the specific Laplacian of the IVT threshold used to identify ARs."

Additionally, in Lines 236-238, we have noted that there is a higher number of ARs originating in the equatorial Atlantic in the reanalysis products compared to the model simulations:

"Despite these outliers occurring at high latitudes, the majority of identified source regions are consistent with atmospheric rivers developing along mid-latitude storm tracks in relation to the baroclinic instability of extratropical cyclones. The reanalyses have more ARs that originate in the equatorial Atlantic compared to the model simulations, though these still represent a small portion that ultimately reach the GrIS (Figure 5)."

**ERA5 in the northernmost part of Greenland. Is it realistic for the other simulations to have ARs there?**

A recent study from Mattingly et al. 2023 investigated extreme melt events in northeast Greenland and linked them to ARs and foehn winds. As such,we have reason to believe that the simulations producing ARs in northeast Greenland are a realistic result. https://doi.org/10.1038/s41467-023-37434-8

> Line 417-418: "Recent studies investigating ARs impacting the northern GrIS support the fact that ARs do occur at such high latitudes in this region (Mattingly et al., 2023)."

**Lines 211 and 220: I think this is wrong. Figure 7b shows the AR area and not the occurrence. The text surrounding Figure 7 needs to be revised based on this.**

Thank you for noticing this. We have significantly revised this section and ensured that the text describing Figure 7 accurately reflects the revised figures and order of presentation. Please see lines 254-270.

**Lines 218-219: This is not necessarily so. It may be that the winds are dropping which will result in smaller IVT and not that precipitation occurred.**

This is a good point and we have revised the text to incorporate this point. For example, we changed "indicating that a large amount of moisture is being transferred" to "suggesting a large moisture transfer from the ARs to the GrIS…" (Line 260-261). We now also mention the potential role of winds dropping in the following statement:

> Lines 272-274: "This increase continues until one day before maximum overlap where all configurations produce a sharp decrease in AR size due to a rapid reduction of moisture and/or winds."

**Line 240: I am missing the link between this sentence and Figure 8. Figure 8 does not show orography, it shows precipitation rate. Please revise this paragraph.**

We agree that the wording of this sentence implies that Figure 8 shows orography. We have revised the text to more clearly explain:

> Line 285-288: "ARs affecting Greenland make landfall on the coast and travel inland. At this point, much of the moisture deposits as precipitation and the storm dissipates. Figure 8 shows the composite precipitation rate of all ARs as they travel over their storm path for one particular grid configuration and remapping scenario. The precipitation rates are largest at the time of maximum overlap with the GrIS, when the storms are at their most inland extent."

**Line 316: "Figure 11 shows that precipitation from ARs likely occur within 500 km of the AR detected by our methods." What is the physical process behind this? If the AR detection point is the core of the AR, then this result may make sense because of possible AR conditions 500 km on either side of the AR core and maximum IVT.**

Thank you for your comment. Section 3.4 - Precipitation has been revised substantially to better describe the physical process (namely the warm front situated ahead of the AR core) behind precipitation patterns within 500 km of the ARs. We have included additional text in the manuscript that gives a more complete description of Figures 9-11. Please see below for further explanation of the 500 km contributing area and its physical processes.

> Line 318-325: "Figures 10 and 11 show snapshots from the models and reanalyses, respectively, of the 95 percentile ARs near the time of their maximum overlap with Greenland, and the outline of the detected feature provided in magenta. The detected feature represents the moist core of the AR, which, unlike the larger synoptic system, does not overlap with a large portion of land at any point throughout its lifecycle (Figure 7d). The snapshots indicate the warm front situated out ahead of the AR core contributes a substantial amount of the storm's precipitation. Additionally large regions of precipitation occur just outside the detected core within the cold front."

> Figure 12a quantifies the impact of including regions outside the core of the AR in compositing precipitation due to that AR."

Reviewer 2:

**In the manuscript "Using variable-resolution grids to model precipitation from atmospheric rivers around the Greenland ice sheet" Waling et al. analyze the impact of different horizontal grid configurations and remapping methods on the simulation of atmospheric rivers that make landfall over Greenland. They find that regionally refined grids produce atmospheric rivers that agree best with state-of-the-art reanalyzes products. The paper is interesting, and the visualization of the data is of high quality. The writing is, in parts, hard to follow and could benefit from careful revisions (I am providing a few more concrete examples below). I listed additional general comments and specific comments below.**

Thank you for your thoughts. We have revised much of the manuscript to provide clarity and reduce its technical nature. We took many of your comments into account (please see below) and believe that our manuscript is improved because of them.

General Comments

1. **I understand that there is a lack of in-situ observations over Greenland but a discussion about the reliability of reanalysis products in this region would be valuable. Specifically, is using ERA5 and MERRA2 precipitation as reference reliable? Are there alternative precipitation estimates available that are more observational-based that could be incorporated into your analysis?**

We chose ERA5 and MERRA2 as our observation-based data as these are the standard datasets used for validation in other studies  (Bresson et al. 2022, Collow et al. 2022, Viceto et al. 2022, Zhou et al. 2022, Mattingly et al. 2023) (Lines 78-79) and an overall lack of ground-based in-situ observations (ie: seven sites in southeast Greenland, records start 2016 onward) on the Greenland Ice Sheet (Box et al. 2023) during our study period (1979-1998). Alternative estimates have recently become available (Box et

al. 2023), but not all products align with our study period (ie: Copernicus Arctic Regional Reanalysis (CARRA) dataset is only available 1991-2021). Additionally, our requirements for observational datasets are constrained by the need to have contemporaneous precipitation rates and IVT in order to detect and track AR features; the reanalysis products that we chose provide these data.

2. **It seems like the phase of precipitation is a significant factor in the mass balance of the ice sheet as you indicated in your introduction. How is the precipitation phase simulated and are there any reliable observations that you could use to validate the model except for reanalysis data that might have its own biases?**

We are not aware of any rainfall datasets for the Greenland Ice sheet that coincide with our simulation period (1979-1998), though newer on-ice data at seven sites in southeast Greenland are becoming available for more recent years, 2016 onward  (Box et al. 2023).

3. **I suggest using more descriptive acronyms for your simulations. E.g., approximate dx and QU, VR, … For instance, instead of using ARCTICGRIS you could use VR_0.125, which would be easier to follow for readers that are not familiar with your grid configurations.**

We appreciate this feedback and agree that choosing more descriptive acronyms with the approximate dx is appropriate. We have made the changes as follows:

f19 —> LL_2°
f09 —> LL_1°
ne30pg2 —> QU_1.5°
ne30pg3 —> QU_1°
ARCTIC —> VR_0.25°
ARCTICGRIS —> VR_0.125°

4. **While I agree with your general conclusion that the representation of topography seems to be the dominant factor in your assessment, the current simulations do not allow you to differentiate those from dynamically better-resolving ARs and their associated processes during landfall. Simulations with high horizontal resolution that have smoothed topography that mimics those in a coarser resolution run would have been insightful to differentiate these two effects. I understand that performing such a simulation is significant work. If you do not have the resources to work on this at the moment, I suggest at least discussing this option for future research in the discussion of your manuscript.**

Thank you for this recommendation. We agree that the most meaningful way forward would be to run our high resolution simulations using the smoothed topography mimicking the coarser resolution runs. Regrettably, we do not have the resources to perform additional simulations as the lead author has since graduated and taken a new position not funded in this area of research. Funding for the lead student author came primarily through the University of New Hampshire as a departmental Teaching Assistant, with smaller contributions from now expired NSF projects. Thus, we take your suggestion to include this in our discussion as future research:

Lines 445-446: "The role of smoothed topography could be further explored by running the model with the VR grid but using the same lower resolution topography as the coarser grids."

We also note that Pollard et al. (2000) and van Kampenhout et al. (2019) have found that simulations using coarse grids precipitate more than observation-based data, and have attributed it to the same phenomena that we have in our study.

Pollard, D., & PMIP Participating Groups. (2000). Comparisons of ice-sheet surface mass budgets from Paleoclimate Modeling Intercomparison Project PMIP simulations. Global and Planetary Change, 24, 79-106.

Van Kampenhout, L., Rhoades, A.M.,Herrington, A.R., Zarzycki, C.M., Lenaerts, J., Sacks, W.J. & Van Den Broeke, M.R. (2019).Regional grid refinement in an Earth system model: impacts on the simulated Greenland surface mass balance. The Cryosphere, 13, 1547–1564.

5.   **What about the vertical grid spacing in climate simulations? You increase the horizontal grid spacing by an order of magnitude. Would a higher vertical grid spacing in the horizontally highly resolved simulations be beneficial?**

We expect that increasing vertical grid spacing could be beneficial, but increasing horizontal grid spacing is of greater concern..

Increasing the vertical resolution would improve the numerical accuracy of the vertical transport and support finer-scale structures such as temperature inversions or cloud macro- and microphysical processes. The CESM3 release will have double the vertical resolution in CAM7. In this co-authors' experience with CAM7, the increased numerical accuracy leads to a less diffusive solution, with stronger gravity waves and vertical transport (see also Skamarock et al. 2019). And while the overall climatology – the large-scale temperature, humidity, clouds and precipitation rates (incl. mountainous regions) – noticeably changed, these changes are smaller than occur due to increasing horizontal resolution. This is because increasing horizontal resolution can support rougher topography boundary conditions and resolves finer-scale resolved processes in GCMs such as grid-scale updrafts (Herrington et al. 2020).

Herrington, A. R., & Reed, K. A. (2020). On resolution sensitivity in the Community Atmosphere Model. Quarterly Journal of the Royal Meteorological Society, 146(733), 3789-3807.

Skamarock, W. C., Snyder, C., Klemp, J. B., & Park, S. H. (2019). Vertical resolution requirements in atmospheric simulation. Monthly Weather Review, 147(7), 2641-2656.

Specific Comments

**Fig. 7b: Does the more rapid decrease in AR area in the reanalysis indicate that the models are not able to extract enough moisture out of the AR once it makes landfall?**

We agree with your observation for ERA5, but less so for MERRA2. The water flux to Greenland from ARs can be understood from Figures 9 and 12, which show that the reanalysis transfers less moisture out of the AR's compared to the coarser grids, which precipitate too much. Although we do not consider nearby ocean points which receive a lot of precipitation as well.

**Fig. 7d: It seems like all simulations largely overestimate the area of overlap. Why is this the case? Are the simulations producing too wide ARs?**

This is likely due to the horizontal resolution of all simulations. The VR configurations are able to resolve the most steep topography of the GrIS, thus allowing the LL and QU grids to penetrate further into the ice sheet. In combination with the actual sizes of ARs seen in Figure 7b, we believe that the VR grids are

indeed producing larger ARs than the reanalyses but think that the larger impacts are due to the resolution of topography.

While model resolution alleviates most of the discrepancy, the larger overlap areas in the VR runs compared to the reanalysis are evident. While this needs to be investigated further, we suspect that using 6-hourly averages instead of 6-hourly instantaneous output for the reanalysis cases does diffuse the metrics (see also dotted purple line in Figure 11b). All else being equal, the magnitude of the Laplacian of the IVT would be larger for instantaneous output, which may result in larger blob areas, and therefore larger AR overlaps areas. However these potential mask size discrepancies are controlled for in Figure 11, where we include regions outside the AR mask in compositing precipitation. That said, this discrepancy between average output in reanalysis and instantaneous output in the models should have been discussed more clearly in the text, and we have done so at Lines 349-351.

**Fig. 8: What is the unit on the color bar? Is this mm/d?**

Thank you for pointing this out. Yes, the units are mm/day. We have added this to the figure and to the caption.

**Fig. 9: The font in this figure is hard to read since it is so small. Additionally, you could consider using a different map projection since the current one results in a lot of white space. Also, it would be helpful to see AR examples from ERA5 and MERRA2 as well.**

Thank you for the feedback. We have increased the font sizes and try to improve readability of the Figure in general. We have included a showing examples of ARs in the reanalyses (Figure 11). We also include the new reanalysis figure below.

[Figure]

**Fig. 10c,d: The differences in total accumulated AR precipitation are concerning. It seems like these differences are coming from an issue in simulating the AR extent rather than its precipitation rate. The strong grid spacing sensitivity indicates that results have not converged yet and that even higher-resolution runs will continue to change the accumulated precipitation over Greenland. Is this a fair statement?**

We agree the results are concerning at coarse resolution. We discuss this finding in lines 439-446. The similarity of the VR_0.25° and VR_0.125° suggest the solutions *are* converging, and therefore addressing their differences with the reanalysis products are the only thing left to determine if these grids are "good enough." We believe this would be important for future work, but is beyond the scope of the current study and experimental design.

**L12: "..., smoothing from coarser resolution latitude-longitude and quasi-uniform grids". It took me a few times to understand what you mean here. I suggest rephrasing this sentence to something like "the coarser resolution latitude-longitude and quasi-uniform grids allow ARS to penetrate further inland due to their smoother topography. "**

We agree. Here is the revised line:

> Lines 14-15: "In contrast, topographic smoothing in coarser resolution latitude-longitude and quasi-uniform grids allows ARs to penetrate further inland on the GrIS."

L13-15: This sentence is hard to understand. I was not able to understand the reason why the VR grid has **lower area-integrated cumulative precipitation and why area-average cumulative precipitation is similar from only reading the abstract.**

We have revised this sentence to hopefully be more clear. Please see:

> Lines 15-17: "Precipitation rates are similar for the VR, latitude-longitude, and quasi-uniform grids, thus leaving the reduced areal extent in VR grids to produce lower area-integrated precipitation."

**L19: Is this true? I thought ARs can also originate in the tropics or sub-tropics.**

ARs can originate in both low- to mid-latitudes; we have revised our statement to include the tropics:

> Line 24: "ARs originate in the low- to mid-latitudes from synoptic scale systems and subsequently travel poleward"

**L41-41: An issue with lat/lon grids is also the odd shape of grid cells in the polar regions, which are high-resolution in the zonal direction but low-resolution in the meridional direction.**

Thank you for this, it is a good point. We have included this sentiment at Lines 48-49: "In addition to this numerical instability, the "stretched" shape of latitude-longitude grids leads to high resolution in the zonal direction but lower in the meridional."

**L85: Something is missing after "...30-second resolution by"**

Given another reviewer's confusion about the role of CLM in the analysis, we have removed this level of detail from the manuscript.

**L119: Why did you use a larger gradient than Ullrich et al. (2021)?**

We wanted to choose a stricter gradient to predict ARs which will be more likely to cause detriment to the GrIS. Please see the following:

The -50k threshold of Patricola et al. (2020) and Rhoades et al. (2020) resulted in too few land-falling storms in Greenland, and we therefore used a more lenient threshold of -30 which produces an order magnitude more land-falling ARs, and is still a larger threshold than used in other TempestExtremes AR studies (Ullrich et al. 2020).

Some outlier source points occur in regions which are inconsistent with conventional definitions of ARs. As noted in the text, our tracker parameters are not as selective as used in other studies, and we have made that choice explicit given the trade-offs between sample size and the strictness of tracker parameters (Lines 137-144).

The attached figure shows the same field but using a more strict threshold for the Laplacian of IVT ($-50,000$ kg m$^{-1}$ s$^{-2}$ rad$^{-2}$). The AR inception points are shifted west and are in closer proximity to the Atlantic Ocean, consistent with the higher IVT threshold for identifying ARs. However, the number of storms dropped significantly, along with the number of storms intersecting Greenland (not shown).

[Figure]

Figure X. Same as Figure 5 in the manuscript, but using stricter TempestExtremes tracking parameters.

Patricola, C.M., O'Brien, J.P., Risser, M.D., Rhoades, A.M., O'Brien, T.A., Ullrich, P.A., Stone, D.A., & Collins, W.D. (2020). Maximizing ENSO as a source of western US hydroclimate predictability. Climate Dynamics, 54, 351-372.

Rhoades, A.M., Jones, A.D., O'Brien, T.A., O'Brien, J.P., Ullrich, P.A., & Zarzycki, C.M. (2020). Influences of North Pacific Ocean Domain Extent on the Western U.S. Winter Hydroclimatology in Variable-Resolution CESM. Journal of Geophysical Research: Atmospheres, 125.

**L195: Fig. 5 misses a closing bracket.**

This has been fixed, thank you.

**L250: I am unsure where to see the 30 mm difference in Fig. 10.**

We have revised this text to make it more clear.

> Lines 291-293: "We used a two-day window centered on the day of maximum AR overlap (Figure 9a) to composite the area-average cumulative AR precipitation (hereafter, precipitation rate), using equation 5. At the end of the two-day window, there is a difference of around 30 mm between the highest and lowest precipitation rates from the grid configurations and reanalyses."

**L308-309:  Ikeda et al. (2010) and Ikeda et al. (2021) would be good additional references here. They show that fairly high-resolution models (down to km-scale) might be needed to resolve complex flow and precipitation interactions with topography.**

Great references to include, thank you. We have included this at Lines 398-399: "Ikeda et al. (2010) and Ikeda et al. (2021) have found similar results describing the high resolution needed to resolve precipitation and flow around steep topography in the western United States."

Ikeda, K., Rasmussen, R., Liu, C., Newman, A., Chen, F., Barlage, M., Gutmann, E., Dudhia, J., Dai, A., Luce, C. and Musselman, K., 2021. Snowfall and snowpack in the Western US as captured by convection-permitting climate simulations: current climate and pseudo global warming future climate. Climate Dynamics, 57(7-8), pp.2191-2215.

Ikeda, K., Rasmussen, R., Liu, C., Gochis, D., Yates, D., Chen, F., Tewari, M., Barlage, M., Dudhia, J., Miller, K. and Arsenault, K., 2010. Simulation of seasonal snowfall over Colorado. Atmospheric Research, 97(4), pp.462-477.

**L322: What would be the reference for such a bias correction?**

We have removed this statement from the manuscript; this statement was not especially well thought out and did not help to describe our work.

**L327: What do you mean with path tracking here and why would it be beneficial?**

We have removed this statement from the manuscript. We originally thought that tracking a storm from its inception to the GrIS could make for an interesting new project. We ended up addressing this point, at least somewhat, with the AR origin figures (Figure 5) that are new since our initial submission.

**L348-350: How about running high-resolution global versions of CEMS2? This might improve ARs in multiple basins and the moisture export into the Arctic in general. Also, how are other models**

**performing in this region? The HighResMIP simulations could be a good opportunity for analysis in future studies.**

This is a good idea for future research, as we do not currently have the computational or funding resources to run high-resolution global versions of CESM2.2.

**L364: There is a question mark in this citation (?Kirbus et al., 2023).**

We have fixed this, thank you.

**L375-376: There is a "that" in the last sentence that should be deleted "We therefore that"**

Great point, this has been fixed.

Reviewer 3:

**This technical paper evaluates the ability of CESM2 to simulate Atmospheric Rivers (ARs) reaching the Greenland ice sheet as well as the sensitivity of the spatial resolution used to model them.**

**The discussion about the impact of the resolution is not original and has already been discussed in Ettema et al. (2009) and Franco et al. (2012) for example. Franco et al. (2012) has exactly the same conclusion than here for example. The ability of CESM2 to simulate ARs is a bit more interesting.**

We appreciate the examples that you have brought forth which further support our findings regarding the impacts of resolution on topography. We have included references to both Ettema et al. (2009) and Franco et al. (2012) in our text (see below). Though that aspect of our study has been discussed already, we believe that our work is of value because we identify the processes responsible for the greater precipitation at coarser resolutions, through looking at individual ARs and their collective behavior, which illustrate greater penetration of storms into the ice sheet interior at coarse resolution.

> Lines 399-402: "Regional modeling studies studies from Ettema et al. (2009) and Franco et al. (2012) forced by reanalysis also found that reduced topographic smoothing at higher resolution simulations improves storm precipitation in Greenland."

**In addition to the justified remarks of the 2 other reviewers, I have 2 additional major remarks before a potential acceptation in WCD.**

1. **Before studying the impact on simulated ARs, the ability of CESM2 + spatial resolution to simulate the mean annual precipitation should be evaluated. As CESM2 is not forced by ERA5, ARs simulated by CESM2 do not occur in the same time than ERA5/MERRA2 and have not the same initial intensity or water content. Therefore, we have differences because ARs are initially not the same ones. As we sometimes say, "apples" are here compared with "pears". How does CESM compare with the mean 1980-1999 annual precipitation? Is there a precip overestimation as for ARs? Do we see the same resolution sensitivity than for ARs? Precipitation from RACMO, MAR or GrSMBMIP should be used as reference for the mean annual precipitation.**

Herrington et al. 2022 evaluated the climatological precipitation and surface mass balance over Greenland in the runs used for this study. There it is shown that mean annual precipitation in the VR grids compare favorably with RACMO products driven by ERAI and ERA5, while the coarser grids precipitated too much over the ice sheet. We will be sure to add this important context to our findings in this study (Line 391-392).

Ensembles are the preferred method of comparing CESM, a free running model constrained only by boundary conditions, to reanalysis. However, CESM is so sensitive to resolution that a single realization can provide robust estimates of climatological variation due to grid and/or dynamical core changes (Herrington et al. 2022). In

Herrington, A. R., Lauritzen, P. H., Lofverstrom, M., Lipscomb, W. H., Gettelman, A., & Taylor, M. A. (2022). Impact of grids and dynamical cores in CESM2. 2 on the surface mass balance of the Greenland Ice Sheet. Journal of Advances in Modeling Earth Systems, 14(11).

> 2. **About the spatial resolution sensitivity experiments, the low resolution topography should be used in the high resolution simulations to confirm that the differences are well due to the ability to resolve the ice sheet topography (the mountain barrier effect) and not to the ability of CESM to resolve precipitation/cloud processes at different spatial resolutions.**

While we agree that this would make for an interesting experiment, we do not have the time or resources to perform further analyses. Funding for the lead student author for this Master's thesis work came primarily through the University of New Hampshire as a departmental Teaching Assistant. Thus, as this is a good suggestion, we have added it to the future research section to pose to other interested scientists doing similar work.

> Line 445-446: "The role of smoothed topography could be further explored by running the model with the VR grid but using the same lower resolution topography as the coarser grids."

Ref:

Ettema, J., M. R. van den Broeke, E. van Meijgaard, W. J. van de Berg, J. L. Bamber, J. E. Box, and R. C. Bales (2009), Higher surface mass balance of the Greenland ice sheet revealed by high-resolution climate modeling, Geophys. Res. Lett., 36, L12501, doi:10.1029/2009GL038110.

Franco, B., Fettweis, X., Lang, C., and Erpicum, M.: Impact of spatial resolution on the modelling of the Greenland ice sheet surface mass balance between 1990–2010, using the regional climate model MAR, The Cryosphere, 6, 695–711, https://doi.org/10.5194/tc-6-695-2012, 2012.

---

## Author Response (AR2)

Dear Dr. Wernli,

We graciously thank the reviewers for their second round of feedback on this manuscript. We have incorporated the recommended changes and believe they have greatly improved the manuscript. In the attached pdf, newly revised text is shown in blue font. We have also included a clean, revised manuscript without markups.

Reviewer comments are shown in **bold**.
Author comments are in plain text.

Sincerely,
A Waling and Coauthors

**Reviewer 1:**

**Thank you to the authors for considering my comments from the first review. I do not have many further comments. Please find them below and I hope they are useful to you.**

Thank you for your further comments. Please see the following text for revisions that we have incorporated into our manuscript.

**Lines 45-46: While I am not overly familiar with the methods used in climate models, the reduction of grid size towards the poles is considered through the use of "reduced grids", such as reduced gaussian grids, in weather forecasting circles (https://confluence.ecmwf.int/display/EMOS/Reduced+Gaussian+Grids). Perhaps some discussion of this, or overlap with climate models, would be good.**

Thank you for this suggestion. We include a brief discussion of reduced gaussian grids and their usage in weather predictions at Lines 46-50:

"Another alternative to traditional latitude-longitude grids common in weather projections (Copernicus, 2019; ECMWF, 2023) is the reduced gaussian grid, which employs quasi-uniformly spaced latitude points and unevenly spaced longitude points to approximate uniform grid size throughout the globe, thus eliminating the need for a polar filter. Though perhaps more efficient than traditional latitude-longitude grids, unstructured grids provide higher computational efficiency and are thus preferred for high resolution global climate models."

**Line 101: Why such an historic period (1 January 1979 to 31 December 1998)?**

1 January 1979 to 31 December 1998 was used in this study because it was the available dataset created by co-author Herrington et al. in 2022. Rather than run new simulations for a two-year Master's thesis, we opted to leverage an existing dataset rather than simulate for a new dataset. Herrington et al. 2022 chose the 1979-1998 time period because that was the default time slice for the Atmospheric Model Intercomparison Project runs in CESM at the time. As 1979 is the beginning of the Satellite period, it is commonly used as a starting point for simulations.

**Figure 4: I agree with you about keeping the boxes for the interquartile range. My suggestion for percentiles was for the whiskers.**

Thank you for re-iterating this point. After further thought, we agree that changing the whiskers to reflect percentiles is the most informative way to show the seasonal data. Please see below for the new figure including 1/99 percentile whiskers.

[Figure]

Figure 4. Number of ARs intersecting the Greenland ice sheet by season, with seasonal peaks in summer and fall. Winter was characterized as December through February, spring as March through May, summer as June through August, and fall as September through November. Seasonal distributions consider 20 years of data (1979-1998) using values from each of the four remapped ensemble members (N=80). Orange line in the center of each box signifies median value and box lower/upper boundaries describe the 25% and 75% quartiles, respectively. The whiskers extend from the box to the 1st and 99th percentiles. Outliers outside these percentiles indicated as open circles.

**Reviewer 2:**

**Thanks for this revised version which improves a lot the quality and interest of the paper. I understand that it is difficult to perform new sensitivity experiments (like using low resolution topography at high resolution) but showing similar figures as Fig2 for the mean precipitation and the 90th percentile precipitation rates simulated by the different configurations using different spatial resolutions will be useful to better highlight the impact of spatial resolution.**

Thank you for this comment. We created plots showing the mean precipitation and 95th percentile precipitation rates without any remapping, please see below. We agree that this conveys useful information and have included the mean precipitation rate figure in our manuscript as Figure 8 with the following text:

Line 261-264: "When we plot annual mean precipitation rate for all model grids and reanalyses on their native grids (Figure 8), the lower resolution grids tend to produce higher precipitation in the interior of the ice sheet, most notably over the southern dome of the GrIS. While the climatological mean precipitation rate is not exclusively from ARs, it exhibits a similar resolution sensitivity to our AR composite precipitation (Figure 7)."

We chose to only include the mean native precipitation rate figure rather than also the 95th percentile rate figure as the spatial trends are extremely similar in both, with the only real difference being the magnitude of precipitation rates; the 95th percentile rates are roughly 5x higher than the mean rates.

Mean native precipitation rates:

[Figure]

95th percentile native precipitation rates:

[Figure]

**In coarser resolution grids, can mean precipitation travel further inland or only maximum (Per 90) can travel?**

Both the mean and maximum precipitation can travel further inland in the coarser grids. As the both the mean (Figure 9a) and 95th percentile (Figure 9b) precipitation rates are similar for all grid configurations, we see the influence of ARs traveling further inland in more smoothed topography in the area-integrated precipitation for both mean and 95th percentile (Figures 9c-d). If this traveling further inland only occurred in 95th percentile ARs, we would not expect to see such a difference among the configurations in mean ARs (Figure 9c). Additionally, the two figures included above describing the mean and 95th percentile native precipitation show similar spatial trends, thus visually supporting this point.

---

## Author Response (AR3)

Dear Dr. Wernli,

We graciously thank you for feedback on our manuscript. We have incorporated the recommended changes and believe they have greatly improved the manuscript. Please refer to the line numbers given below to see changes in the revised, unmarked version of the manuscript.

Reviewer comments are shown in **bold**.
Author comments are in plain text.

Sincerely,
A Waling and Coauthors

**Dear Annelise**

**We are almost there ... just a few minor points for you to consider before the paper is finally accepted (the 2nd one is important!):**

**- L49: I don't understand the logic of this newly introduced sentence: "Though ... more efficient, unstructured grids provide higher ... efficiency ...". This does not make sense to me and I don't understand what "efficient" means in the first part of the sentence vs. the 2nd part. Please correct this sentence or omit it.**

Thank you for this note, we agree that this sentence was unclear and have removed it.

**- Fig. 5: the first sentence of the caption is hard to read / understand. I think "eventually" is not needed. To me it would read much better as "Origin of summer ARs that intersect the GrIS in JJA." Also the next sentence is most likely not proper English (or at least very complicated). Just write something like "The size and color of the dots indicate ..." --> please check all figure captions and improve readability where possible!!**

We have revised many of the figure captions for readability, please see below.

Figure 1: We added Latitude-Longitude (LL), Quasi-Uniform (QU), and Variable-Resolution (VR) labels to each of the three columns for clarity.

[revised manuscript text omitted]

**With best regards,**
**Heini**